# Phage homing endonuclease amplifies anti-defense genes to evade bacterial immunity

Kotaro Chihara[1], Aa Haeruman Azam[1], Artyom A. Egorov[2], Ilya Terenin[2], Masanori Hashino[3], Koichi Watashi [1], Kazuhiro Horiba [3], Vasili Hauryliuk [2] & Kotaro Kiga[1] ✉

Bacteriophages must overcome diverse bacterial immune systems, yet the molecular principles enabling such escape remain poorly understood. Here, we show that the phage homing endonuclease SegB facilitates immune evasion by promoting the segmental amplification of anti-defense loci. The anti-phage defense Septu inhibits phage T6 replication by cleaving the variable loop of tRNA$^{Tyr}$. We show that SegB enables immune evasion by amplifying a genomic segment that contains the full-length *tRNA$^{Tyr}$* gene. This repeat expansion increases tRNA$^{Tyr}$ expression, allowing the phage to overcome Septu immunity. SegB also mediates in trans amplification of distinct anti-defense genes that counteract OLD and toxin-antitoxin ToxIN defense systems. Collectively, our findings demonstrate that SegB-mediated segmental amplification represents a versatile mechanism by which phages rapidly adapt to and circumvent diverse bacterial antiphage defenses.

Bacteria have evolved diverse antiphage defense systems to combat bacteriophage invasions. Beyond CRISPR-Cas and restriction-modification, recent genome-wide surveys have broadened the understanding of these systems through bioinformatic predictions that identify candidates adjacent to known defense systems[1–3], mining of prophage genomes[4], and random cloning of DNA fragments from strain collections[5]. Collectively, these studies demonstrate that mobile genetic elements (MGEs) are major drivers of the evolution of antiviral immunity in prokaryotes. MGEs contribute to the formation of defense islands—genomic diversity hotspots encoding antiphage defense systems—and, in doing so, benefit both the host and the MGE itself[6–8]. These systems are frequently mobilized between bacterial genomes via MGEs, thus enabling the continuous exchange of defense genes that are essential for bacterial survival[9].

In turn, phages employ various anti-defense systems that either directly inhibit individual defense proteins or restore depleted host resources, thereby enabling defense evasion[10,11]. Like defense genes, anti-defense genes often co-localize, forming anti-defense islands[12]. However, it remains unclear whether MGEs contribute to the mobilization and dissemination of anti-defense islands in phage genomes, and if so, what molecular mechanisms shape this process.

Homing endonuclease genes (HEGs) represent a widespread class of minimalistic MGEs encoded in phage genomes[13]. To propagate, HEGs cleave specific target sites in alleles lacking the HEG, thereby recruiting host-mediated homologous recombination to insert themselves into the broken chromosome. Through this process, known as homing, HEGs can spread efficiently within phage populations, frequently accumulating near essential genes. Beyond their selfish mobility, HEGs also influence phage genome evolution by inducing double-strand breaks (DSBs) during co-infection, thereby promoting horizontal gene transfer, gene shuffling, and adaptation[14,15]. Furthermore, recent studies have shown that some HEGs enhance viral competitiveness by impairing rival phages[16]. Yet, despite their prevalence and potential impact on phage genome dynamics, the extent to which HEGs shape phage interactions with host defense systems remains largely unexplored.

Here, we reveal that the homing endonuclease SegB, a MGE encoded by phage genomes, amplifies anti-defense genes to

[1]Department of Drug Development, National Institute of Infectious Diseases, Japan Institute for Health Security, Tokyo, Japan. [2]Department of Experimental Medical Science, Lund University, Lund, Sweden. [3]Pathogen Genomics Center, National Institute of Infectious Diseases, Japan Institute for Health Security, Tokyo, Japan. ✉e-mail: kiga.k@jihs.go.jp

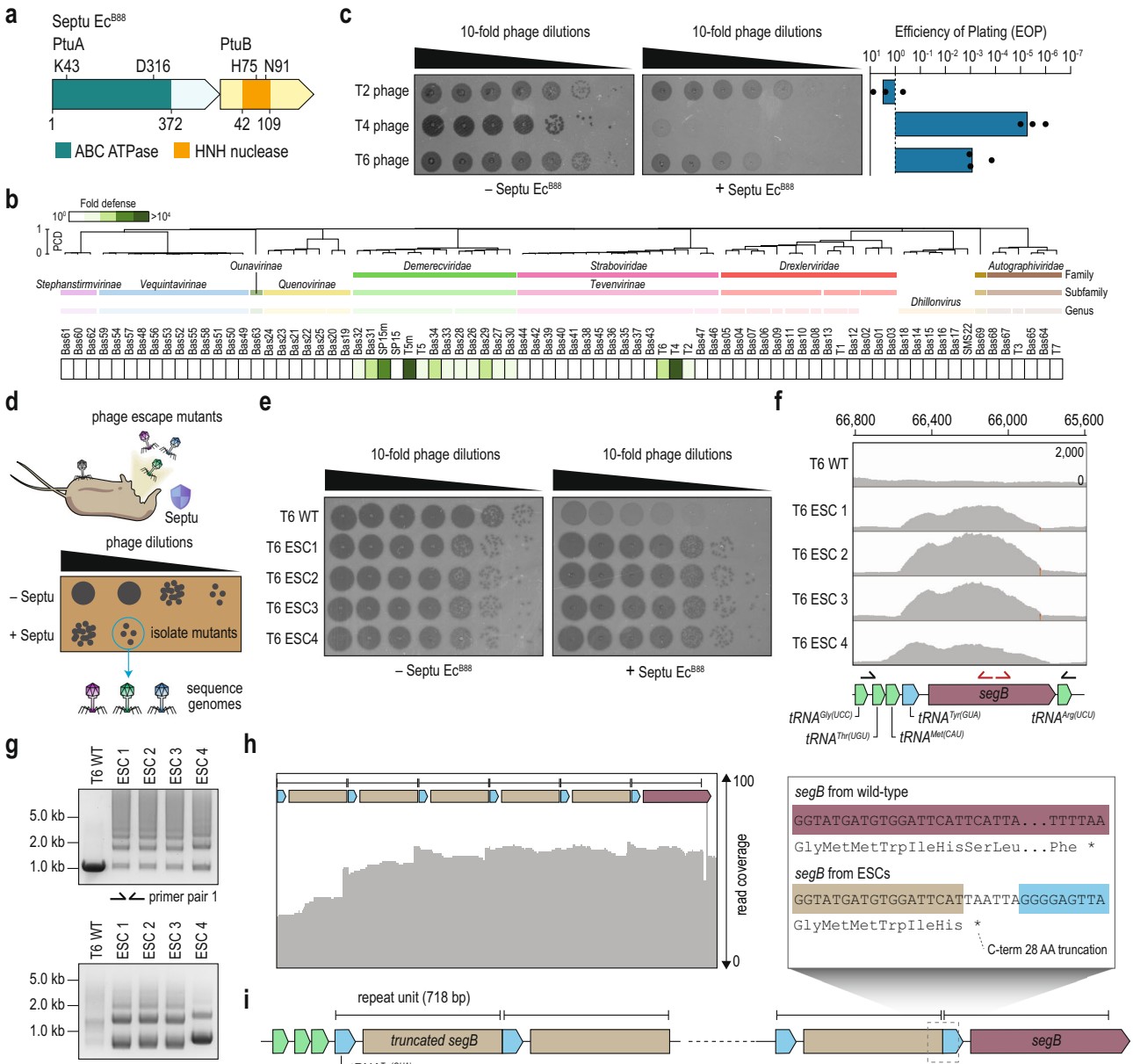

**Fig. 1 | T6 phage mutants evade Septu Ec^BB8 immunity via segmental amplification of the tRNA–SegB locus. a** Genetic architectures of Septu Ec^BB8 encoded by *Escherichia coli* JBBDAGI-19-0041. The ABC ATPase domain and the HNH nuclease domain are indicated in green and orange, respectively. **b** Defense profile of Septu Ec^BB8 against BASEL phage collections and common laboratory coliphages. The phage order and dendrogram are determined through hierarchical clustering based on the proteome composition distance matrix (see "Methods"). **c** A plaque assay of T-even phages on cells with and without Septu Ec^BB8. The efficiency of plating is shown next to the plaque image. Shown is the average of three technical replicates, with individual data points overlaid, and no statistical test was performed. **d** A schematic representation of the isolation of Septu Ec^BB8 escaper (ESC) phages. Wild-type phages infected cells containing Septu Ec^BB8. The diluted, cleared lysate was applied onto a lawn of Septu Ec^BB8. Four distinct plaques were isolated, and their genomes were sequenced. **e** A plaque assay of wild-type and ESC T6

phages that evade Septu Ec^BB8 immunity, on cells with and without Septu Ec^BB8. **f** Coverage profiles for the regions encoding tRNAs and SegB from both wild-type and ESC T6 phages. The corresponding locus map is shown below the profile. Arrows indicate the primer pairs utilized in (**g**) for PCR analysis. Genomic coordinates are based on the Enterobacteria phage T6 complete genome (GenBank accession no. AP018814.1). **g** Gel electrophoresis results for the PCR products obtained with the primer pairs shown in (**f**). The gel shown is representative of *n* = 3 technical replicates from a single biological sample. Coverage profile for the assembled contig from PacBio long-read sequencing (**h**) and a detailed view (**i**). The corresponding locus map is shown above the profile. A schematic highlights a 718 bp repeat unit containing *tRNA^Tyr* and a C-terminal truncated *segB*. The sequence of the boundary for each repeat unit is shown in the enlarged box. Source data are provided as a Source data file.

evade multiple bacterial immunity systems. We investigate this mechanism in the context of the Septu defense from an *Escherichia coli* clinical strain (hereafter Septu Ec^BB8), a system comprising *ptuA* and *ptuB*, also present in the group I-A retron-Eco7 defense system (Fig. 1a and Supplementary Fig. 1a, b)[17,18]. The current study was motivated by our previous findings that retron-

Eco7 inhibits phage replication via host tRNA^Tyr cleavage during infection, and that T5-like phages can evade this immunity by encoding tRNA^Tyr[19,20]. Continuing our studies of this tRNA-centered arms race, here we dissect the mechanisms of Septu Ec^BB8-mediated defense and viral SegB-mediated counter-defense.

## Results

### Septu Ec[B88] provides antiphage immunity

First, we investigated the protective effects of Septu Ec[B88]. A DNA fragment spanning the coding sequence and the 300 bp flanking regions was amplified from the source genome and cloned into a pLG001 empty vector (p15A origin, expressed from the native promoter)[2]. We transformed *E. coli* DH10B, which naturally lacks Septu, with the Septu-expressing plasmid or empty vector, and challenged the bacteria with various phages (Fig. 1b). Like Septu from *E. coli* ATCC 25922 and retron-Eco7, which both effectively block phage T5[19,20], Septu Ec[B88] primarily inhibited phages of the Demerecviridae family. Furthermore, Septu Ec[B88] inhibited T-even phages with varying efficacies (Fig. 1c). While phage T2 was resistant to Septu Ec[B88], the system caused a significant reduction in the efficiency of T4 plating; T6 exhibited intermediate sensitivity. To ascertain whether the defensive activity of Septu Ec[B88] depends on predicted ABC ATPase and HNH nuclease activities, we constructed mutants of the Walker A/B (K43 or D316) motifs of PtuA as well as the H-N-H motif (H75 or N91) of PtuB. Substitutions targeting either motif abolished immunity, establishing that both enzymatic activities are essential for defense (Supplementary Fig. 1c).

To probe whether Septu Ec[B88] functions as an abortive infection system, we infected bacteria expressing PtuAB with increasing multiplicities of infection (MOI) of T4 and T6 phages in liquid culture. While cells lacking Septu Ec[B88] were quickly lysed, cells expressing PtuAB were not; however, bacterial growth was retarded at high MOIs (Supplementary Fig. 2a). Furthermore, the quantity of phage progeny was markedly reduced in the presence of the system, as evidenced by the ratio of phage titers in the supernatant before and after infection (Supplementary Fig. 2b). The decrease in PFU ratio at high MOI in the absence of Septu Ec[B88] likely reflects lysis from without, a known outcome of high multiplicity adsorption in T-even phages, and is independent of Septu activity. Thus, we concluded that Septu Ec[B88] impairs the growth of phage-infected cells, disrupting the phage replication cycle, which is typical of abortive infection[21]. However, during T6 infection at high MOI, cells expressing functional PtuAB were lysed, and the number of released phages increased, even in the presence of the defense system (Supplementary Fig. 2c, d). These observations suggest that while Septu Ec[B88] limits phage propagation under certain conditions, it is less effective at high MOI. Given the moderate protection against T6 (Fig. 1b, c), these findings imply that phage T6 may carry a gene dosage-sensitive evasion mechanism that overcomes the Septu defense when multiple phage particles co-infect the same cell.

### Septu-resistant T6 mutants amplify *tRNA[Tyr]* and *segB* genes

Isolating spontaneous escape mutants that evade antiphage immunity, followed by genome sequencing, sheds light on the nature of phage triggers that activate immunity or phage countermeasures that inhibit it[22]. Therefore, we attempted to isolate spontaneous escape mutants of phages capable of circumventing Septu-mediated immunity (Fig. 1d). We successfully isolated four independent T6 mutants resistant to Septu Ec[B88] (Fig. 1e). Whole-genome sequencing of the mutants revealed no shared mutations, including single-nucleotide polymorphisms (SNPs) or small insertions/deletions (indels), among all escape phages (Supplementary Fig. 3a). Two of these mutants exhibited a common non-synonymous mutation within a gene encoding a tail sheath (KMB99_gp234). However, the co-expression of Septu Ec[B88] and the tail sheath did not result in bacterial cell death, suggesting that the tail sheath gene is not responsible for triggering Septu Ec[B88] (Supplementary Fig. 3b). In addition to SNPs and indels, we observed an increase in read coverage within the locus encoding tRNAs and SegB homing endonucleases (Fig. 1f). This may indicate either the presence of multiple copies distributed throughout the viral genome or local tandem repeats. To distinguish between these two possibilities, we performed PCR amplification on the locus using either flanking primers or inverse primers. PCR with flanking primer pairs produced a ladder of products in the escape mutants, whereas a single band was observed in the wild-type. PCR with inverse primer pairs yielded bands in the escape mutants but not the wild-type, indicating that segmental amplification resulted in the formation of tandem repeats (Fig. 1g).

We then quantified the tandem repeats in the genomic DNA using PacBio HiFi long-read sequencing. Due to the presence of glucosyl-5-hydroxymethylcytosine modifications in the genome[23], long-read sequencing yielded limited reads from each mutant. Therefore, we combined the reads from all the mutants into a single FASTQ file before assembly (Supplementary Fig. 3c). This approach produced two contigs: one encompassing the T6 genome variant sequence and another approximately 4500 bp in length. We found that both contigs contained a 718-bp repeat unit comprising *tRNA[Tyr]* and *segB* (Fig. 1h, i and Supplementary Fig. 3d). Within this single repeat, a 28-amino-acid truncation was identified at the C-terminus of SegB (Fig. 1i). Additionally, we observed microhomology between the 5′ end of *tRNA[Tyr]* and the 3′ end of *segB*, which may facilitate homologous recombination and remove the 3′ termini of the *segB* sequences (Supplementary Fig. 3e). Collectively, our results demonstrated that Septu-evading T6 phages acquired tandem repeats of a genomic region encoding tRNA and a C-terminally truncated version of SegB.

### Septu Ec[B88] cleaves tRNA[Tyr] at the variable loop

Given that the locus amplified in the Septu-escaping phages contains multiple tRNA genes (Supplementary Fig. 4a), we initially considered whether any of the genes within this locus, not limited to tRNA[Tyr], could be responsible for neutralizing the immunity of Septu Ec[B88] (Fig. 2a). We co-expressed each gene with Septu Ec[B88] and infected the cells with phage T6. Overexpression of tRNA[Tyr] rescued viral infection under Septu Ec[B88] immunity (Fig. 2b). The rescue of T6 infection was analogous to the results in our previous study, wherein the immunity of retron-Eco7 was neutralized by supplementation with T5 tRNA[Tyr] [19]. In the earlier report, we determined that retron-Eco7 cleaved the variable loop of *E. coli* tRNA[Tyr], thus interfering with phage propagation; conversely, T5-derived tRNA[Tyr] was resistant to cleavage due to its distinctive variable loop[19,24]. Northern blotting confirmed that intact *E. coli* tRNA[Tyr] levels were diminished in cells expressing Septu Ec[B88] under T6 infection (Fig. 2c). Furthermore, the exogenous phage-derived tRNA[Tyr] can partially suppress Septu-mediated degradation of host tRNA[Tyr] under overexpression conditions in a manner consistent with its ability to weaken Septu immunity and restore phage replication (Supplementary Fig. 4b and Fig. 2b). In addition to testing the overexpression conditions, we constructed T6 mutants with a deletion in the tRNA-rich region (TRR) and confirmed that the ΔTRR mutant showed enhanced sensitivity to the immunity provided by Septu Ec[B88] compared to the wild-type (Fig. 2d). Consistent with this model, phage T4, which encodes SegB but lacks a tRNA[Tyr] gene, was unable to escape Septu immunity (Supplementary Fig. 4a). Collectively, our findings support the idea that increased expression levels of tRNA[Tyr] are crucial for neutralizing defense (Fig. 2e).

Subsequently, to precisely delineate the Septu-mediated tRNA cleavage site, we quantified the 5′ terminus of tRNA fragments through tRNA fragment sequencing. A pronounced peak of read coverage from the 5′ end of the cleavage site was observed within the variable loop of the *E. coli* tRNA[Tyr] from cells expressing Septu Ec[B88] during T6 infection, whereas no peaks were detected in the control cells (Fig. 2f). Moreover, the absence of peaks without T6 infection substantiated that viral infection initiated the cleavage of *E. coli* tRNA[Tyr] via Septu Ec[B88]. Analysis of the read coverage ratio between cells with and without Septu Ec[B88] pinpointed the cytosine at position 48 and adenine at position 49 within the stem of the variable loop as the cleavage sites (Fig. 2g). In contrast, multiple peaks throughout the anticodon and variable loops

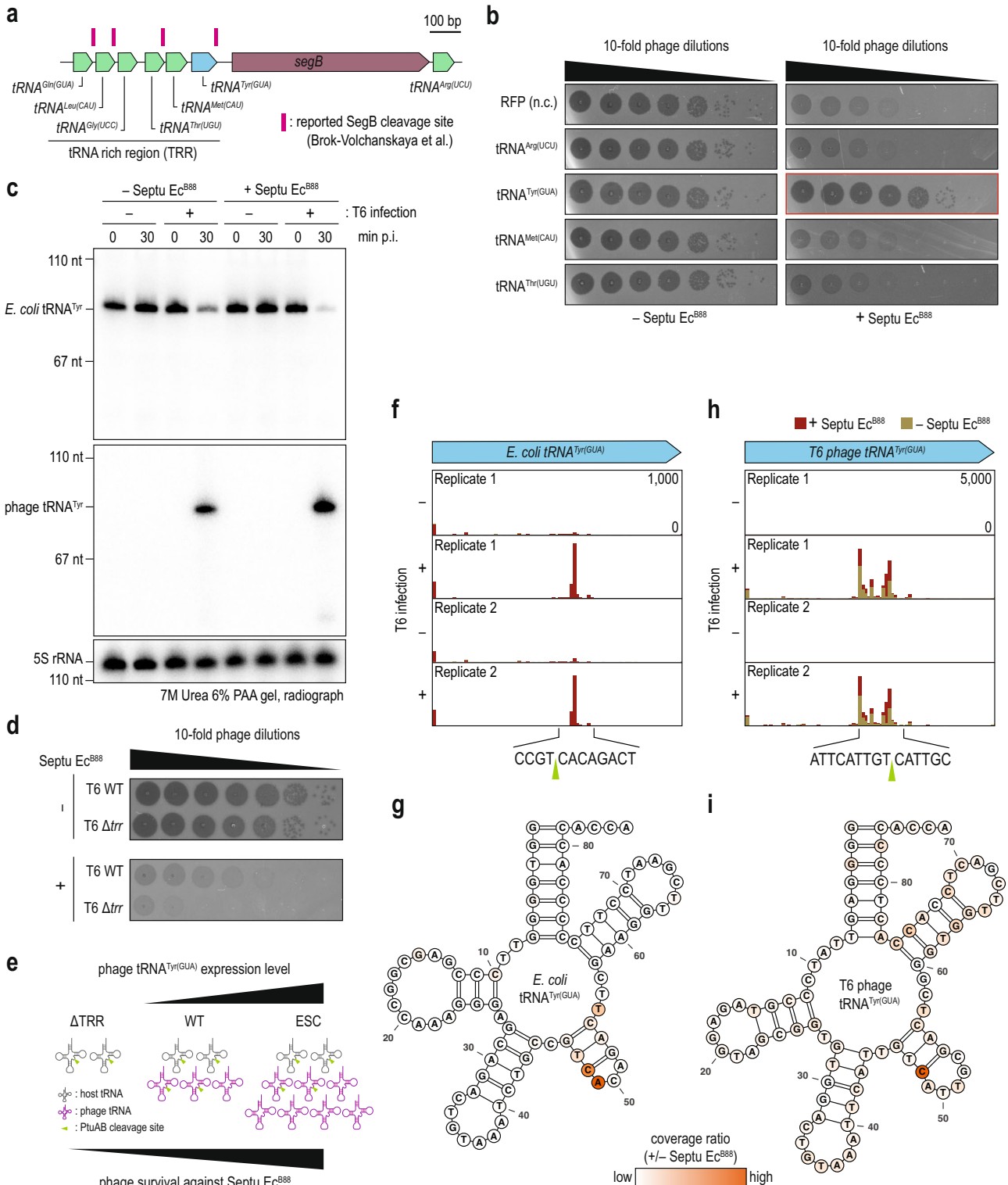

**Fig. 2 | Septu Ec^B88 cleaves the variable loop of the tRNA^Tyr. a** tRNA-rich region (TRR) adjacent to *segB* gene in the T6 phage genome. Potential SegB cleavage sites reported by Brok-Volchanskaya et al.[26] are represented by pink lines. **b** Plaque assay of T6 phage on cells co-expressing Septu Ec^B88 with the RFP control, tRNA^Arg, tRNA^Tyr, tRNA^Met, and tRNA^Thr from T6 phage. **c** Northern blot analysis of the *E. coli* and T6 phage tRNA^Tyr upon phage infection. Total RNAs were resolved by 7 M Urea 6% polyacrylamide gel. 5S rRNA is used as a standard. p.i., post infection. The blot shown is representative of *n* = 2 biologically independent experiments performed using independently extracted RNA samples. **d** A plaque assay of T6 phage wild-type and the Δ*trr* mutant on cells expressing an empty vector or Septu Ec^B88.

**e** Relationship between T6 phage survival against Septu Ec^B88 immunity and expression levels of phage tRNA^Tyr. Coverage of the 5′ base of uniquely aligned reads obtained from tRNA-fragment-seq. The regions encoding (**f**) *E. coli* tRNA^Tyr and (**h**) T6 phage tRNA^Tyr are highlighted. In each replication, the coverages in the presence and absence of Septu Ec^B88 are overlaid. Potential cleavage sites are indicated by green triangles. Coverage ratios comparing the presence and absence of Septu Ec^B88 are depicted on the secondary structures of (**g**) *E. coli* tRNA^Tyr and (**i**) T6 phage tRNA^Tyr. The secondary structures of tRNA^Tyr were depicted using VARNA[61].

suggest that T6 tRNA$^{Tyr}$ was relatively unstable compared to *E. coli* tRNA$^{Tyr}$ (Fig. 2h). Nevertheless, quantifying the read coverage ratio between cells with and without Septu Ec$^{B88}$, we identified a single site of differential enrichment, indicating that the cytosine at position 48 in T6 tRNA$^{Tyr}$ was the cleavage site (Fig. 2i). Together, these results suggest that Septu Ec$^{B88}$ indiscriminately degrades both host- and phage-encoded tRNA$^{Tyr}$ species.

## SegB functionality is essential for segmental amplification of phage *tRNA$^{Tyr}$*

In addition to the increased copy number of *tRNA$^{Tyr}$*, which contributes directly to neutralizing the Septu-mediated defense, we hypothesized that the adjacent *segB* gene, also amplified, promotes segmental amplification through its activity as a homing endonuclease. Homing endonucleases are either intron-encoded or free-standing enzymes that promote propagation into homologous alleles lacking endonucleases[13,25]. A previous study suggested that SegB may cleave within tRNA-containing sequences, but its functional role in phage biology has remained unexplored[26].

To determine the role of SegB in segmental amplification of *tRNA$^{Tyr}$*, we conducted experimental evolution via serial passaging with a T6 Δ*segB* mutant and the isogenic wild-type phage. *E. coli* cultures were initially infected at low multiplicity of infection (MOI ~ 0.1). After overnight incubation, supernatants were collected and used to infect fresh *E. coli* cultures, and this cycle was repeated five times. Thereafter, the loci subjected to segmental amplification were PCR-amplified and analyzed via gel electrophoresis. In evolved wild-type T6, the locus encoding SegB was segmentally amplified, producing ladder-like products under pressure from Septu Ec$^{B88}$ immunity (Fig. 3a, *left*). In contrast, the Δ*segB* mutant did not produce these products after serial rounds of infections, underscoring the essentiality of SegB for segmental amplification (Fig. 3a, *middle*). To test whether SegB's nuclease activity is essential for this process, we generated a catalytic-dead mutant SegB$^{Y17A/G19A}$. A T6 phage strain encoding SegB$^{Y17A/G19A}$ similarly failed to segmentally amplify the locus during the serial passaging experiment, thereby phenocopying the Δ*segB* mutant (Fig. 3a, *right*). These results demonstrate that the nuclease activity of SegB is indispensable for segmental amplification during phage evolution in vivo.

To assess whether SegB directly cleaves the region that undergoes amplification, we performed in vitro cleavage assays using a ~2-kb DNA fragment encompassing the *segB*-proximal locus (Supplementary Fig. 5a, b). Purified SegB showed detectable cleavage activity in the presence of Mg$^{2+}$ alone, but the reaction was markedly enhanced upon addition of Mn$^{2+}$, yielding pronounced dose-dependent cleavage products (Fig. 3b). Catalytically dead SegB$^{Y17A/G19A}$ and SegB$^{Y17F/G19A}$ variants showed no detectable activity under the same conditions (Supplementary Fig. 5c). Given the endonucleolytic activity of SegB, an elevated copy number of *segB*, despite truncation of its C-terminus, could have adverse effects on the fitness of both phage and its bacterial host. To assess this, we measured the growth of *E. coli* upon overexpression of either wild-type SegB, its C-terminal-truncated variant (hereafter SegB$^{short}$), catalytically dead variants SegB$^{Y17A/G19A}$ and SegB$^{Y17F/G19A}$, and tRNA$^{Tyr}$ (Supplementary Fig. 5d, e). As expected from its nuclease activity, overexpression of wild-type SegB caused a strong growth inhibition, whereas no growth defect was detected upon overexpression of neither SegB$^{short}$, catalytically dead variants, nor tRNA$^{Tyr}$. Furthermore, serial passaging experiments with Septu-escaper phages showed that ladder-like bands from segmentally amplified sequences gradually decreased during successive infections in cells lacking Septu Ec$^{B88}$ immunity, with the population converging to a predominant product of approximately the wild-type size (Fig. 3c and Supplementary Fig. 5f). Although we did not sequence the collapsed product, these data indicate that the segmental amplifications are unstable in the absence of immune pressure. Collectively, our findings suggest that SegB is required for the induction of DSBs

followed by amplification of genes critical for counteracting antiphage defense mechanisms.

To further elucidate how SegB selects its genomic targets for cleavage and subsequent amplification, we mapped the SegB-dependent DNA breaks genome-wide. To do so, we used purified SegB and high-molecular-weight phage genomic DNA to perform the selective enrichment and identification of tagged genomic DNA ends by sequencing (SITE-seq)—a method that was originally established for the mapping of Cas9 cleavage specificity[27]—and then we mapped the locations of SegB-mediated breaks by peak calling (Fig. 3d; see "Methods"). This analysis revealed 83 reproducible cleavage sites across the phage genome (Fig. 3e and Supplementary Data 1). As the genomic region surrounding *segB* and the adjacent tRNA array is central to SegB-mediated escape, we focused on the read coverage at this locus. In good agreement with previous reports[26], this analysis revealed multiple sharp 5′ end peaks within the tRNA genes themselves (Fig. 3f). Furthermore, motif analysis of the ±20 bp surrounding all cleavage sites using MEME identified a conserved sequence motif RAWTSGAAC (23/83 sites; E-value: $5.6 \times 10^{-6}$) (Fig. 3g). To directly validate the functionality of the identified motif biochemically, we generated dsDNA substrates either containing or lacking the identified sequence and subjected them to in vitro cleavage assays. Quantification of cleavage efficiency revealed that substrates harboring the consensus motif exhibited a substantially higher SegB-dependent cleavage than motif-disrupted variants, although the latter still produced detectable cleavage products (Fig. 3h). These findings indicate that the identified motif enhances SegB-mediated cleavage but is not strictly required for DNA processing in vitro, raising the possibility that sequence preferences may influence, but do not solely determine, the genomic regions susceptible to SegB-dependent amplification in vivo.

Finally, we assessed the importance of homologous recombination-related genes downstream of the SegB-mediated DSBs. According to an established model of T4 recombination, once the 3′ end of single-stranded DNA (ssDNA) was resected, the phage-encoded ssDNA-binding protein Gp32 and recombination mediator UvsY sequestered the ssDNA, forming a tripartite complex and priming it for recruitment of the phage-derived RecA-like protein UvsX (Supplementary Fig. 6a)[25]. Notably, we utilized *E. coli* DH10B carrying a loss-of-function *recA1* allele, thereby exhibiting a markedly low efficiency of homologous recombination. Experimental evolution via serial passaging showed that although the locus encoding SegB from the wild-type T6 showed segmental amplification, Δ*uvsX* and Δ*uvsY* mutants produced no ladder-like product after five successive infections (Supplementary Fig. 6b, c). These results indicate that the segmental amplification of *segB* and *tRNA$^{Tyr}$* is facilitated by homologous recombination.

## SegB homing endonuclease is frequently associated with tRNAs

Thus far, we have demonstrated that SegB drives segmental amplification of tRNAs, a process critical for neutralizing the tRNA-targeting antiphage defense Septu in phage T6 (Figs. 2 and 3). Next, we asked whether SegB-mediated amplification of tRNA genes is widespread across various phages by searching for additional instances of *segB* co-localization with tRNAs. First, we built a Hidden Markov Model profile for SegB derived from phage T6 using clustered representatives obtained from the BLASTp results (see "Methods"). We analyzed the conservation of SegB across all bacteriophage and archaeal virus proteins available in the GenBank database and identified a total of 1832 hits. Subsequently, we generated a phylogenetic tree using 386 non-redundant SegB homologs with ≤95% identity and ≥40% coverage (Fig. 4a and Supplementary Data 2). Phylogenetic analysis revealed that SegB homologs are encoded by phages infecting various phyla, including Pseudomonadota, Bacillota, Bacteroidota, and Cyanobacteriota. In some cases, SegB from Pseudomonadota-infecting

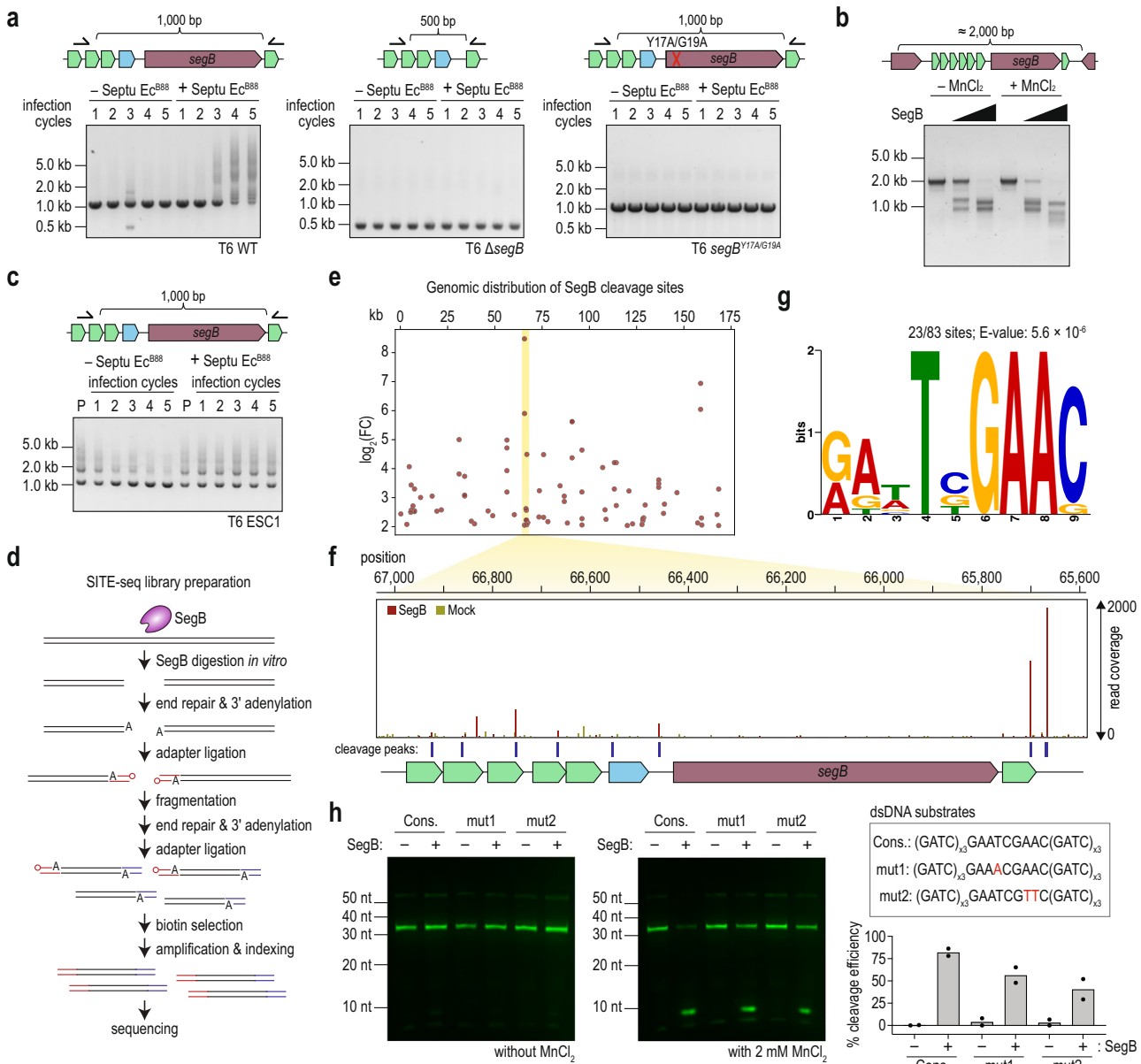

**Fig. 3 | SegB is essential for amplifying the tRNA^Tyr gene. a** Gel electrophoresis results for the PCR products obtained from T6 phage wild-type, ΔsegB, or segB^Y17A/G19A lysates in the experimental evolution via serial passaging. Primer pairs are indicated on the corresponding locus map above. **b** Nuclease activity assay of SegB against a PCR product containing the SegB-encoding TRR locus. All reactions contained 1 mM MgCl$_2$, which supports basal SegB activity. MnCl$_2$ was added where indicated to test the reported Mn$^{2+}$ preference of SegB. The PCR fragment used in the assay is indicated in the locus map above. **c** Gel electrophoresis result for the PCR products derived from the lysate of Septu Ec^B88 ESC mutants in the experimental evolution via serial passaging. **d** Schematic overview of the SITE-seq workflow used to identify SegB cleavage sites. HMW phage gDNA is incubated with purified SegB in vitro. Following end-repair and 3' adenylation, a 5' biotinylated adapter is ligated to the DNA. The DNA is then enzymatically fragmented, end-repaired, 3'-adenylated, and ligated to a 3' adapter. Biotinylated fragments are enriched, PCR-amplified, indexed, and subjected to high-throughput sequencing. **e** Genomic distribution of SegB cleavage sites. Peak midpoints were calculated as the average of the start and end coordinates, converted to kilobases, and plotted against their corresponding log$_2$FC(SegB-treated/Mock-treated). A log$_2$FC > 2 filter was applied prior to visualization. Reproducible cleavage sites were defined as sites detected in at least two out of three SITE-seq replicates. **f** Coverage profiles of SITE-seq reads for the regions encoding tRNAs and SegB under SegB-treated and mock conditions. The corresponding locus map is shown below the profile. Blue lines mark the peaks identified by our peak-calling procedure (see Methods). Genomic coordinates are based on the Enterobacteria phage T6 genome (GenBank accession no. AP018814.1). **g** MEME motif analysis of sequences spanning ±20 bp around all SegB cleavage peaks. Numbers indicate the counts of peaks containing the predicted motifs. **h** Nuclease activity assay of SegB using dsDNA substrates containing the identified motif. All reactions included 1 mM MgCl$_2$, which supports basal SegB activity, and MnCl$_2$ was added where indicated to assess the reported Mn$^{2+}$ preference of SegB. The sequences of the DNA substrates are shown to the right of the gel images. Cleavage efficiency was calculated using ImageJ as the percentage of cleaved products relative to total DNA under Mn$^{2+}$-containing conditions. Cons., consensus motif. The gels shown are representative of n = 2 technical replicates from a single biological sample. Source data are provided as a Source data file.

phages was transferred to phages infecting Cyanobacteriota, supporting the idea that SegB can be horizontally transferred between phage genomes infecting different bacterial phyla. SegB is also frequently encoded by phages of the Myovirus morphotype, although

this pattern may partially reflect database sampling bias. Nonetheless, the observed distribution is consistent with the idea that SegB has been repeatedly domesticated and co-opted by Myoviruses despite its mobile origin.

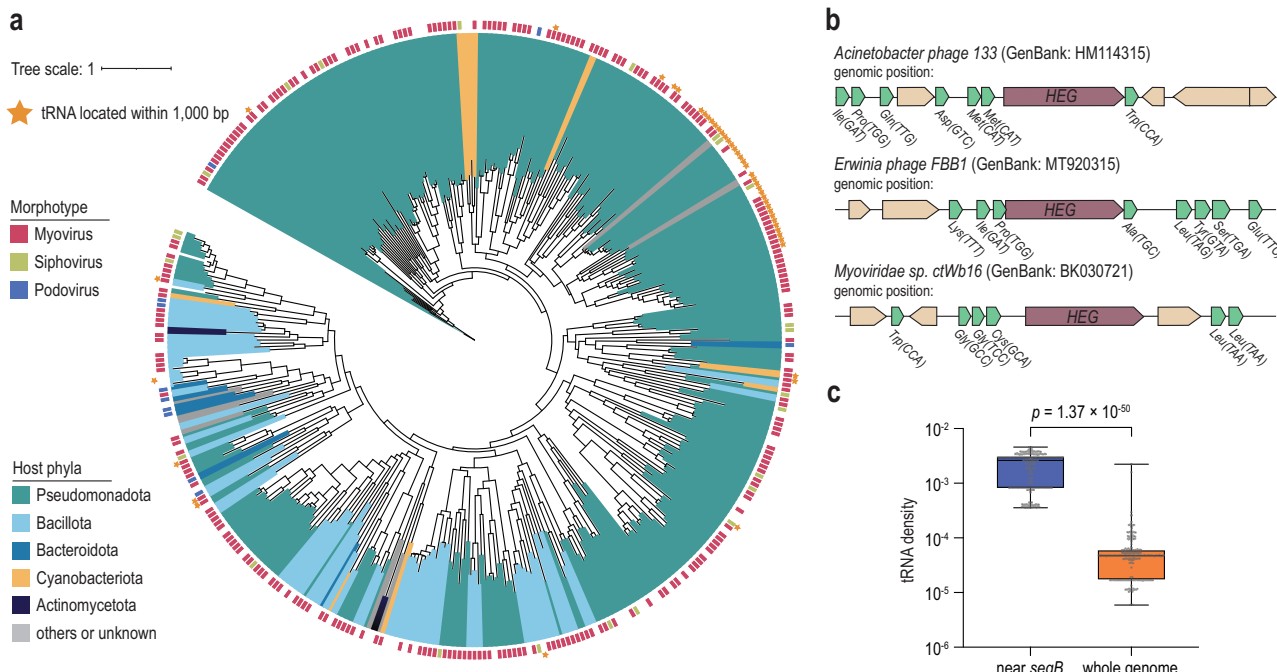

**Fig. 4 | SegB is frequently associated with tRNAs in phage genomes.**
**a** Phylogenetic tree of phage-encoded SegB homologs. All non-redundant sequences with a 95% identity threshold were used. The outer circle indicates the morphotype of phages corresponding to each SegB homolog. SegB homologs located within 1000 bp upstream or downstream of tRNA are marked with a star. **b** Representative examples of SegB and tRNA co-localization. The names of the phages and the accession numbers of the relevant genomes in the GenBank database are indicated at the top. HEG, homing endonuclease gene. **c** Boxplots showing the distribution of tRNA gene density within 1000 bp upstream or downstream of

*segB* genes and across the entire genomes. Each dot represents an individual phage genome ($n = 296$). For each genome, tRNA density in the *segB*-proximal region was paired with the genome-wide tRNA density. The centre line indicates the median, the box represents the 25th–75th percentiles, and the whiskers extend to the most extreme data points within 1.5× the interquartile range. A one-sided paired Wilcoxon signed-rank test was performed to assess differences in tRNA density between two regions (alternative hypothesis: segB-proximal >genome-wide; statistic = 43956.0, $p = 1.37 \times 10^{-50}$).

We then analyzed gene neighborhoods of the 386 non-redundant *segB* genes retained within the phylogenetic tree (Fig. 4a). tRNAscan-SE[28] revealed multiple instances of co-localization of tRNAs and *segB* within ±1000 bp in phage genomes (Fig. 4a, b, Supplementary Fig. 7a, and Supplementary Data 3). Notably, such co-localization events were enriched in phages infecting Pseudomonadota. Comparison of tRNA densities near *segB* genes with genome-wide tRNA densities indicated that tRNAs were significantly co-localized with *segB* (Wilcoxon signed-rank test, $p = 1.37 \times 10^{-50}$) (Fig. 4c and Supplementary Fig. 7b). This pattern did not appear to favor particular tRNA isotypes, as a wide range of isotypes was observed (Supplementary Fig. 7c). In summary, phylogenetic and gene neighborhood analyses highlighted a widespread association between *segB* and tRNAs.

## SegB drives evasion of the antiphage defenses OLD and ToxIN

The observed widespread co-localization of *segB* and tRNA genes across multiple phage genomes prompted us to investigate whether SegB mediates segmental amplification of additional tRNAs against other potential tRNA-targeting antiphage defense systems[19,29,30]. To this end, we first tested four tRNA-targeting defense systems—Retron Eco7, PrrC, PARIS, and OLD—for their ability to restrict T6. While Retron Eco7 and PrrC did not inhibit T6 replication, both PARIS and OLD provided strong defense. We therefore focused on these two systems and performed serial passaging experiments to examine whether SegB enables escape under these conditions (Supplementary Fig. 8a). Among the tested systems, only wild-type T6, and not its Δ*segB* derivative, formed clear plaques upon successive rounds of infection of bacteria expressing OLD (Supplementary Fig. 8b). OLD is a single polypeptide defense system composed of ABC ATPase and TOPRIM domains, proposed to prevent lambda phage superinfection

through tRNA degradation and inhibition of protein synthesis[30,31]. Surprisingly, no segmental amplification was observed at the *tRNA–segB* region (Supplementary Fig. 8c). This prompted us to further investigate how T6 evades the OLD defense system in a SegB-dependent manner. To identify the genomic alterations responsible for evading OLD, we isolated four T6 mutants that overcame OLD-mediated immunity (Fig. 5a). Whole-genome sequencing of these mutants revealed an increase in read coverage within the loci encoding Gp49.1–NrdC.1, ~22 kb distant from the *segB* gene (Fig. 5b). Like Septu-resistant phages, we identified ladder-like products from OLD-resistant phages by flanking PCR amplification. Furthermore, the inverse primer pairs yielded bands from the OLD-resistant mutants but not the wild-type (Fig. 5c). These results indicate that OLD-resistant phages segmentally amplify the *gp49.1–nrdC.1* region. Experimental evolution of wild-type T6 and its Δ*segB* mutant via serial passaging using T6 wild-type and Δ*segB* mutant confirmed that the segmental amplification of the locus is SegB-dependent, producing ladder-like products under OLD immune pressure (Fig. 5d); Δ*uvsX* and Δ*uvsY* mutants yielded analogous results (Supplementary Fig. 8d, e). To determine whether SegB directly targets the OLD-associated amplified locus, we performed in vitro cleavage assays using a 2-kb DNA fragment encompassing the amplified region. Purified SegB cleaved this substrate in a metal-dependent manner, albeit less efficiently than the tRNA-containing locus (Fig. 5e). Thus, SegB-mediated DSBs, followed by homologous recombination, are essential for the segmental amplification of the relevant locus and OLD defense evasion. Moreover, our results imply that even though the anti-defense factors are not located adjacent to the *segB* gene, SegB can amplify anti-defense genes in trans elsewhere in the viral genome.

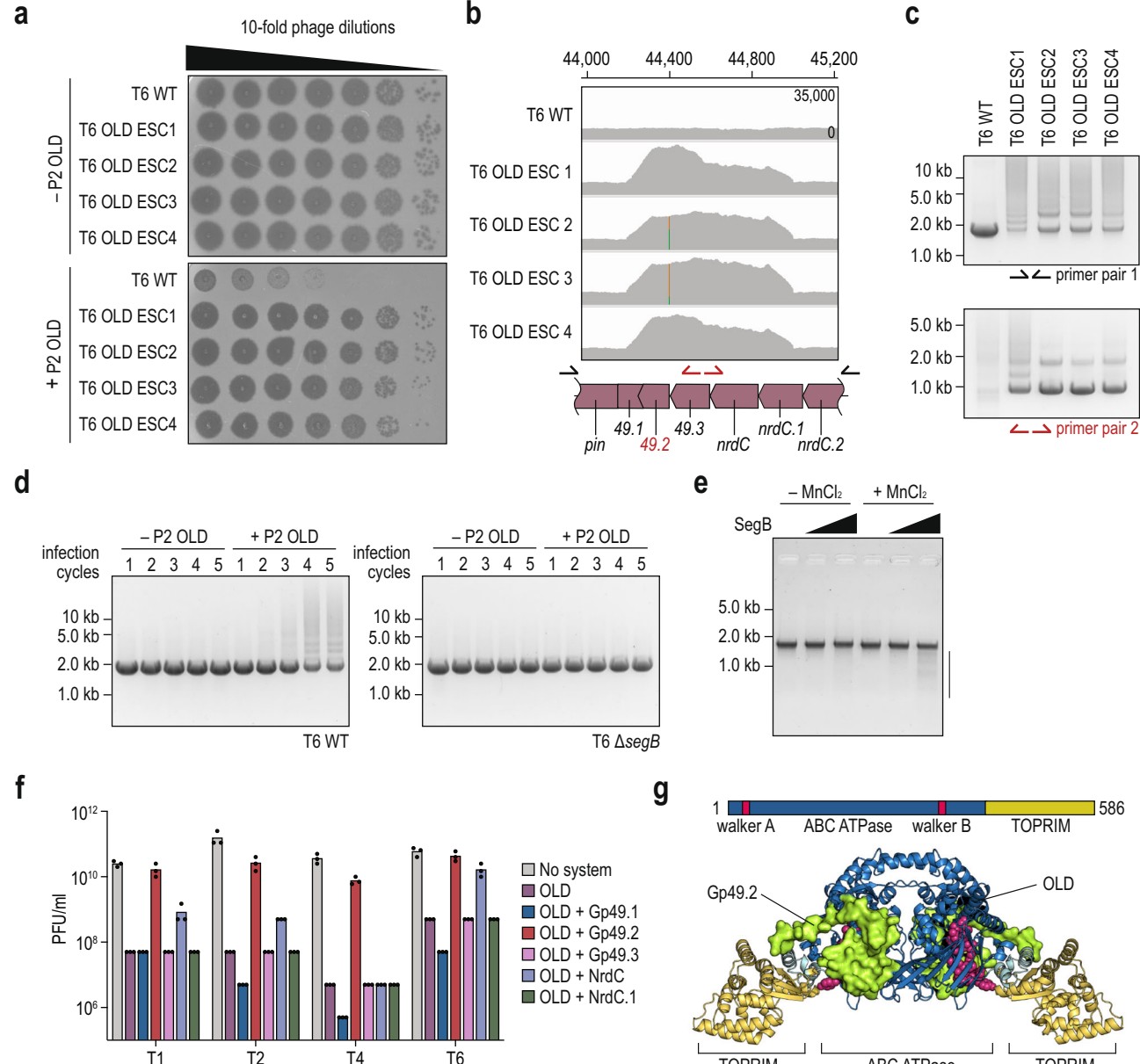

**Fig. 5 | SegB is capable of in trans amplification of anti-defense genes to counteract the OLD defense system. a** A plaque assay of T6 phage wild-type and OLD escapers (ESC) that evade OLD immunity, using cells expressing either an empty vector or OLD defense. **b** Coverage profiles for the regions encoding Pin to NrdC.2 from T6 wild-type and OLD ESC phages. The corresponding locus map is shown below the profile. Arrows indicate the primer pairs utilized in (**c**) for PCR analysis. Genomic coordinates are based on the Enterobacteria phage T6 complete genome (GenBank accession no. AP018814.1). **c** Gel electrophoresis results for the PCR products obtained with the primer pairs shown in (**b**). **d** Gel electrophoresis results for the PCR products obtained from T6 phage wild-type or Δ*segB* lysates in the experimental evolution via serial passaging. Primer pair 1, depicted in (**b**), is utilized for PCR of the relevant locus. **e** Nuclease activity assay of SegB against a PCR product containing the *pin-nrdC.2* locus. All reactions contained 1 mM MgCl₂,

which supports basal SegB activity. MnCl₂ was added where indicated to test the reported $Mn^{2+}$ preference of SegB. The PCR fragment used in the assay is shown in (**b**). The gels shown are representative of $n = 2$ technical replicates from a single biological sample. **f** Plaque-forming units of T1, T2, T4 and T6 phages on cells co-expressing OLD with the RFP control, Gp49.1, Gp49.2, Gp49.3, NrdC, and NrdC.1 from T6 phage. Shown is the average of three technical replicates, with individual data points overlaid, and no statistical test was performed. **g** Tetrameric complex of P2 OLD and Gp49.2 as predicted by AlphaFold3 (ipTM = 0.55)[34]. The prediction was based on dimerized OLD using the crystal structure of *Thermus scotoductus* (PDB: 6P74) as a ref. [71]. Walker A/B motifs are represented as magenta spheres. Schematics of the domain organization are shown above the structure. Source data are provided as a Source data file.

We then investigated the involvement of the segmentally amplified region in evading OLD immunity. We independently expressed five genes in the presence of OLD and subjected the cells to phage infection. While Gp49.1, Gp49.3, NrdC, and NrdC.1 had no effect on OLD anti-phage activity, Gp49.2 abolished the defense phenotype, indicating that Gp49.2 functions as a phage-encoded inhibitor of OLD (Fig. 5f). HHpred[32] suggested a potential similarity between Gp49.2 and the

antitoxin RelB3 from *Methanocaldococcus jannaschii* (probability: 79.12, *E*-value: 8.8)[33]. Moreover, phylogenetic analysis of Gp49.2 indicates that Gp49.2 is exclusively conserved in Straboviridae infecting bacteria of the Pseudomonadota phylum (Supplementary Fig. 9a, b and Supplementary Data 4). To gain insight into Gp49.2 function, we cofolded P2 OLD and Gp49.2 using AlphaFold3[34]. AlphaFold3 yielded medium-confidence predictions for the interaction (ipTM = 0.55) and suggested

that Gp49.2 occludes the Walker A/B motifs in the ATPase domain, likely inhibiting the ATPase activity of OLD (Fig. 5g). To experimentally test the functional relevance of the predicted OLD–Gp49.2 interaction, we performed structure-guided mutagenesis of Gp49.2. AlphaFold3 predictions suggested that Gp49.2 engages the ATPase domain of OLD through a small surface involving residues M13, T15, and W32 (Supplementary Fig. 9c). We therefore generated single (M13A, T15A, W32A) and triple (M13A/T15A/W32A) alanine substitutions targeting this putative interaction interface. Whereas the T15A and W32A substitutions had little to no effect on Gp49.2-mediated inhibition of OLD defense, the M13A substitution strongly reduced protection against OLD in T2 and T6, while retaining partial inhibitory activity in the context of T4. In contrast, the triple mutant fully lost inhibitory activity (Supplementary Fig. 9d). These results identify M13 as a critical determinant of Gp49.2 anti-defense function and provide functional support for a specific interaction surface between Gp49.2 and OLD predicted by AlphaFold3. Overall, our additional screening of SegB-mediated evasion of antiphage defense systems highlights the ability of SegB to segmentally amplify anti-defense factors both in cis and in trans.

Finally, building on the observation that SegB can mediate in trans segmental amplification of *gp49.2* under OLD immunity, we investigated whether SegB also contributes to a previously documented amplification-based evasion strategy. ToxIN, a type III toxin-antitoxin system, mediates antiphage defense via an abortive infection mechanism[35]. To counter ToxIN, phage T4 encodes an anti-ToxIN factor TifA which undergoes segmental amplification under ToxIN immunity[36]. We performed experimental evolution via serial passaging using a Δ*segB* T4 mutant. As previously observed, the TifA locus in wild-type T4 was segmentally amplified under ToxIN pressure, producing ladder-like products. Strikingly, the Δ*segB* mutant failed to generate such products even after five infection cycles (Supplementary Fig. 10). Although Δ*segB* phages showed no detectable amplification of the *tifA* locus, escapers occasionally appeared in the spot assay (Supplementary Fig. 10c). These likely arise from SegB-independent mutations that prevent ToxIN activation, explaining the mismatch between the spot and PCR results. Taken together, our findings suggest that SegB-mediated segmental amplification may represent a shared strategy within T-even phages, although further studies will be required to determine how broadly this mechanism is deployed across more distantly related phages.

## Discussion

Our study identifies a homing endonuclease-driven strategy that enables phages to evade multiple, mechanistically distinct bacterial defense systems through segmental amplification of anti-defense genes. The GIY-YIG family endonuclease SegB promotes amplification both in cis (tRNA^Tyr against Septu) and in trans, acting on loci located tens of kilobases away, such as *gp49.2* (~22 kb from *segB*) under OLD immunity and *tifA* (~52 kb from *segB*) under ToxIN immunity, thereby providing a dosage-based route to neutralize immunity.

Phage genomes often encode tRNAs, yet their physiological roles remain incompletely understood[37,38]. The prevailing view is that phage-derived tRNAs help compensate for mismatches between phage codon usage and the host tRNA pool, and recent work has shown that such compensation can facilitate efficient translation during infection in Cyanobacteria[39]. Another proposed function is to replenish host tRNAs that become depleted as part of antiviral responses. Because Septu Ec^B88 degrades tRNA^Tyr—shared by both *E. coli* and T6 phages—phage-encoded tRNA^Tyr may act as a decoy or supplement during infection. Indeed, T5 circumvents tRNA-targeting defenses by rapidly expressing its own tRNAs[19,29], whereas T6 relies on SegB-mediated amplification to increase tRNA^Tyr dosage. These contrasting strategies illustrate that phages have evolved multiple solutions to maintain translation in the face of tRNA-targeting defenses, with SegB providing a distinct, recombination-based route to boost tRNA levels when required.

SegB is a GIY-YIG family homing endonuclease implicated in introducing DSBs within tRNA-containing sequences[26]. In our system, segmental amplification is likely initiated by SegB-mediated cleavage, followed by homologous recombination between the 5′ region of the tRNA^Tyr gene and the 3′ region of the *segB* gene, which share micro-homologous sequences. Notably, SegB is a free-standing homing endonuclease, encoded independently of introns or inteins, that can target DNA sites far from its own locus. Unlike intron-encoded homing endonucleases, which typically cut within ~25 bp of their insertion site[40], free-standing endonucleases require extensive DNA resection and homologous recombination, enabling co-conversion of the endonuclease gene and distal DNA segments[41]. This likely enables the *trans* amplification of the anti-defense gene *gp49.2* under selective pressure from the OLD defense system, with SegB-induced DSBs triggering long-range homologous recombination to increase anti-defense gene dosage. In addition to SegB, T-even phages encode several GIY-YIG family homing endonucleases, including I-TevI and I-TevII, which are both associated with group I introns[40–46]. Furthermore, phage T4 encodes five HNH family HEGs, MobA–E, which introduce single-strand nicks into their target DNA, generating recombinogenic intermediates[46,47]. Whether these endonucleases also contribute to the segmental amplification of anti-defense factors remains largely unknown and merits further study.

How SegB avoids destroying the phage genome despite its nuclease activity remains an important open question. T-even phages possess a dedicated recombination-dependent replication system driven by UvsX, UvsY, Gp32, and host RecBCD, which likely repairs SegB-induced DSBs through strand invasion and recombinational restart[48,49]. Such a robust repair framework would enable the phage to tolerate sporadic endogenous breaks. Furthermore, our SITE-seq analysis shows that SegB does not cleave the genome indiscriminately, and strong cleavage sites share recognizable sequence features, suggesting that cleavage of the wild-type locus is intrinsically limited rather than widespread (Fig. 3e–h). Prior studies of free-standing GIY–YIG endonucleases have demonstrated that cutting of the donor allele is typically inefficient, whereas robust cleavage is directed toward recipient alleles lacking the endonuclease gene, providing a natural mechanism for avoiding lethal self-targeting[50]. Additionally, phage DNA-binding proteins may occlude potential target sites. In this context, the phage T4 protein MotB has been proposed to markedly compact and restructure the nucleoid, making it plausible that MotB-coated DNA physically limits homing endonucleases from accessing their target sequences[51]. Together, these considerations suggest that SegB activity is modulated by a combination of repair capacity, sequence preference, and structural or chromatin-like constraints, allowing the phage to balance the selfish mobility of SegB with preservation of genome integrity.

The reversible genomic amplifications we observe in phage T6 closely parallel the "genomic accordion" model described for poxviruses, in which virus genomes transiently expand and contract in response to selective pressures[52]. Studies in Vaccinia virus and, more recently, Monkeypox virus have demonstrated that adaptive gene amplifications can arise rapidly during host–virus conflict and collapse once the pressure is relieved, underscoring genome plasticity as a general strategy for viral evolution[52,53]. Our findings suggest that phages can employ an analogous principle: SegB-mediated amplifications that provide a rapid but unstable route to overcome antiphage defenses. This conceptual connection places phage adaptive genome dynamics within a broader framework shared across diverse large DNA viruses.

Collectively, our results reveal a previously unrecognized role for free-standing homing endonucleases as active participants in the phage-host arms race. Beyond selfish propagation, they can couple DNA cleavage to segmental amplification, providing a generalizable and rapid-response strategy to overcome diverse bacterial immunity systems.

## Methods

### Bacteria, phages, and growth media

*E. coli* K-12 DH10B, MC1061, and *E. coli* clinical isolates distributed by Antimicrobial Resistance Research Center, National Institute of Infectious Diseases were streaked on Luria-Bertani (LB) agar plates and grown overnight at 37 °C. Overnight cultures were prepared at 37 °C in LB medium with shaking at 200 rpm. Antibiotics were used as needed at concentrations listed as follows: 100 µg/ml carbenicillin, 50 µg/ml kanamycin, and 20 µg/ml chloramphenicol. Arabinose and glucose were used as needed at concentrations of 0.2%, respectively. All phages used in this study are listed in Supplementary Data 5.

### Plasmid construction

All plasmids and oligonucleotides are listed in Supplementary Data 6, 7, respectively. The DNA fragments encoding Septu Ec[B88], P2 OLD, and ToxIN, encompassing the coding DNA sequences (CDSs) along with the 300 bp upstream and downstream, were amplified from the source genomes. Only the CDS for PARIS was synthesized (Supplementary Data 8). The amplified sequences were cloned into inversely amplified pLG001 PCR products using NEBuilder HiFi DNA assembly (NEB, #E2621). The pLG001 vector (p15A origin, chloramphenicol-resistant) was kindly provided by Feng Zhang (Addgene plasmid #157879). The coding DNA sequences of phage proteins were amplified from the purified phage genome. The amplified sequences were then cloned into inversely amplified pKLC83ara (pBR322 origin, ampicillin-resistant) PCR products under the control of the arabinose-inducible promoter using NEBuilder HiFi DNA assembly (NEB, #E2621).

### Phage infection assay

**Plaque Assay.** A bacterial lawn for the spot assay was prepared by combining 300 µl overnight bacterial culture with 10 ml of LB soft agar (LB, 0.5% agarose gel, 1 mM CaCl₂), followed by layering this mixture onto 1.5% LB agar with the requisite antibiotics. Subsequently, a ten-fold serial dilution of the phage lysate was prepared using sodium magnesium (SM) buffer. Two microliters of the phage dilution was applied to the bacterial lawn, which was then incubated at 37 °C overnight. Imaging of the plaques was conducted using an EPSON flatbed scanner model GT-X980. Each plaque assay was performed in biological triplicate.

**Liquid Infection Assay.** Overnight cultures were subjected to a 100-fold dilution in fresh LB medium, followed by the transfer of 200 µl of the diluted aliquot into a 96-well plate. Subsequently, 2 µl of a serially diluted phage lysate was applied to each well to achieve a predetermined MOI. The plate was then sealed with a gas-permeable Breathe-Easy® sealing membrane (Diversified Biotech BEM-1, #Z380059) and incubated within a LogPhase 600 Microbiology Reader (Agilent Technologies). The optical density at 600 nm (OD₆₀₀) was monitored every 10 min over a 12-h period, with shaking conducted at 600 rpm at 37 °C. Each liquid infection assay was performed in technical triplicate.

### Toxicity assay

The toxicity assay with overexpression of phage proteins was conducted in accordance with previously established protocols[19]. Briefly, a ten-fold serial dilution of overnight bacterial cultures was generated in LB medium. 4 µl of each aliquot was applied to LB plates that contained the requisite antibiotics and supplemented with either 0.2% arabinose or 0.2% glucose. The plates were subsequently incubated overnight at 37 °C. Spot images were captured utilizing an EPSON flatbed scanner GT-X980. The images presented in the figures are representative of three independent biological replicates.

### Isolation of escape mutants

Overnight cultures of bacteria expressing Septu Ec[B88] were diluted 100-fold in fresh LB medium supplemented with 20 µg/ml of chloramphenicol and 1 mM CaCl₂. Following 1 h of incubation at 200 rpm and 37 °C, phages were added into the culture at an MOI of 1 and subsequently incubated overnight under the same conditions. The cultures were then subjected to centrifugation at 5000 × g for 10 min, after which the supernatant was filtered through a 0.45 µm filter. A ten-fold serial dilution of the filtrate was prepared, and 2 µl of each dilution was spotted onto a bacterial lawn expressing Septu Ec[B88] and incubated at 37 °C overnight. The following day, four distinct plaques were isolated from the plates and propagated by incubating them with bacteria expressing Septu Ec[B88] in 2 ml of LB medium supplemented with 20 µg/ ml of chloramphenicol and 1 mM CaCl₂ at 37 °C with shaking at 200 rpm for 3 h. The lysate was centrifuged at 5000 × g for 10 min, and the supernatant was filtered through a 0.45 µm filter to eliminate any residual bacteria. To verify that the phages obtained could evade the Septu immunity, a bacterial lawn suitable for the spot assay was prepared by mixing 300 µl of an overnight bacterial culture containing either an empty vector or plasmids encoding Septu Ec[B88] with 10 ml of LB soft agar. This was then layered onto 1.5% LB agar containing 20 µg/ ml of chloramphenicol. Ten-fold serial dilutions of the potential escape mutant phages and wild-type phages were prepared, and 2 µl of each dilution was spotted onto the bacterial lawns and incubated at 37 °C overnight.

### DNA extraction from escaper phages

Cultures of bacteria expressing Septu Ec[B88] were incubated overnight and subsequently diluted 100-fold in fresh LB medium supplemented with 20 µg/ml of chloramphenicol and 1 mM CaCl₂. Following 1 h of incubation at 200 rpm and 37 °C, phages were added to the culture at an MOI of 1, followed by an additional overnight incubation under the same conditions. The resulting cultures were then subjected to centrifugation at 5000 × g for 10 min, and the supernatant was filtered using a 0.45 µm filter. The supernatant was treated with DNase I (Takara, Japan) and RNase A (Takara, Japan) at final concentrations of 3 U/ml and 20 ng/ml, respectively, and was incubated at 37 °C for 1 h. Subsequently, PEG 8000 and NaCl were added to the filtered supernatant at final concentrations of 10% and 4%, respectively, and the mixture was allowed to remain at 4 °C overnight. On the following day, the mixture was centrifuged at 10,000 × g for 1 h, with the supernatant being discarded. The resulting phage pellet was resuspended in 500 µl of SM buffer. This suspension was mixed with 12 µl of 10% (w/v) SDS and 12 µl of 0.5 M EDTA (pH 8.0) and incubated for 10 min at 68 °C. Subsequently, 12 µl of 3 M sodium acetate (pH 6.5) was added. Thereafter, 600 µl of phenol:chloroform:isoamyl alcohol (25:24:1, P:C:I) was added, and the mixture was vortexed for 30 s. The tube was centrifuged for 3 min at 16,000 × g. The aqueous phase was transferred to a new tube, and 600 µl of chloroform:isoamyl alcohol (24:1, C:I) was added. After vortexing for another 30 s, the mixture was centrifuged again for 3 min at 16,000 × g, with the C:I extraction being repeated twice. The aqueous phase was subsequently precipitated with 2 volumes of ice-cold ethanol, 1/30 volume of 3 M sodium acetate (pH 6.5), and 1 µl of GlycoBlue™ coprecipitant (Invitrogen, #AM9516), for 2 h at −30 °C. The precipitated pellet was washed with ice-cold 75% ethanol, briefly dried at 20 °C for 5 min, and finally resuspended in 50 µl of sterilized water.

### DNA-seq data analysis

DNA sequencing was conducted either with DNB-seq in paired-end mode (2 × 150 bp) at BGI Genomics Co. or NovaSeq 6000 in paired-end mode (2 × 150 bp) at Azenta Life Sciences. Raw sequence reads of the wild-type T6 phage were assembled using Shovill version 1.1.0 with default settings (https://github.com/tseemann/shovill). Gene annotations were performed using Pharokka version 1.3.2 on the assembled genomes, also with default parameters[54]. Utilizing the assembled genomes as references, single-nucleotide polymorphisms (SNPs) were identified from the raw sequence reads using Snippy version 4.6.0 with

default settings (https://github.com/tseemann/snippy). Large genomic deletions or insertions were manually examined using the generated BAM files and visualized with Integrative Genome Viewer version 2.16.0[55].

## DNA gel electrophoresis

PCR-amplified products from either extracted phage DNAs or crude phage lysates were mixed with 10× DNA loading dye (TAKARA). Equal amounts of the DNA samples were separated using 1% agarose gel in 1× TAE and 5% Midori Green Extra (Nippon Genetics). The gels were then visualized under a Blue-Green LED light at a wavelength of 500 nm. Imaging of the gels was conducted using FAS-Digi PRO (Nippon Genetics).

## PacBio long-read sequencing and the data analysis

Long-read sequencing was performed on a PacBio Revio at Macrogen, Inc. The raw reads from four ESC mutants were merged into a single FASTQ file, and Flye was used for assembling the raw reads in−pacbio-hifi mode[56], which produced two contigs. Each contig was indexed, and the raw reads were aligned using bwa mem in pacbio mode. The resulting sam file was converted to a bam file, which was sorted and indexed with Samtools[57]. The resulting mapping data was visualized using Integrative Genome Viewer version 2.16.0[55].

## RNA extraction

Cultures of bacteria expressing Septu Ec[B88], which were grown overnight, were diluted 100-fold in fresh LB medium with 20 μg/ml of chloramphenicol and 1 mM $CaCl_2$. Upon reaching $OD_{600}$ of 0.5, phages were added into the culture at an MOI of 10. After either 15 min or 30 min post-infection, a stop solution (95% v/v ethanol and 5% v/v water-saturated phenol (pH >7.0)) was added to the culture and then immediately stored at −80 °C. The frozen culture was defrosted on ice and pelleted for 20 min at 5000 × $g$ and 4 °C. Total RNA was extracted using the hot phenol method. Briefly, the pellet was resuspended in 600 μl TE buffer, 0.5 mg/ml lysozyme (pH 8.0), and 60 μl 10% (w/v) SDS, then heated for 5 min at 65 °C before adding 66 μl NaOAc (pH 5.2). Next, 750 μl phenol was added and mixed every 30 s for 5 min at 65 °C, followed by centrifugation for 15 min at 16,000 × $g$ and 15 °C. The aqueous phase was transferred to a phase-lock gel tube, and 750 μl chloroform was added. After mixing, the mixture was centrifuged for 10 min at 16,000 × $g$ and 15 °C. The aqueous phase was precipitated with 2 volumes of ice-cold ethanol and 1/30 volume of 3 M NaOAc (pH 6.5) for 2 h at −20 °C. The pellet was washed with ice-cold 75% ethanol and resuspended in 40 μl sterilized water. Then, 10 μl DNase solution (2 units of TURBO DNase™ and 10 units of SUPERase·In™ RNase Inhibitor in DNase buffer) was added, incubated for 30 min at 37 °C, followed by adding 100 μl sterilized water. Finally, 150 μl of the mixture was transferred to a phase-lock gel tube with 150 μl phenol:chloroform:isoamyl alcohol, mixed, and centrifuged for 15 min at 16,000 × $g$ and 15 °C. The aqueous phase was precipitated again with 2.5 volumes of ice-cold ethanol and 1/30 volume of 3 M NaOAc (pH 6.5) for 2 h at −20 °C. The pellet was washed with 75% ethanol and resuspended in 40 μl sterilized water.

## RNA gel electrophoresis and northern blotting

10 μg of the extracted RNAs were mixed with 2× RNA loading dye and subjected to heat denaturation at 95 °C for 3 min. Following this, the RNA samples were placed on ice for 5 min and subsequently separated using 6% denaturing PAGE in 1× TBE and 7 M urea. The gels were then transferred to Hybond+ membranes (GE Healthcare) and subjected to UV crosslinking with 120 mJ of UV light at a wavelength of 254 nm. The UV crosslinked membranes were then probed with tRNA-specific, radioactively labeled DNA oligonucleotides in PerfectHyb Plus (Sigma, cat# H7033) at 42 °C overnight. The probed membranes were washed every 15 min with 5× Saline Sodium Citrate (SSC)/0.1% SDS, 1× SSC/0.1% SDS, and 0.5× SSC/0.1% SDS buffers at 42 °C. Autoradiograms were visualized using Typhoon FLA 7000 (GE Healthcare) and quantified employing ImageJ.

## cDNA library preparation and sequencing

cDNA library preparation followed by sequencing was conducted by Filgen, Inc. and Arraystar Inc. Initial assessments of total RNA samples for integrity were performed using agarose gel electrophoresis. RNA concentrations, alongside protein contamination, were quantitatively measured using a NanoDrop ND-1000 spectrophotometer (Thermo Fisher Scientific); RNA quality was subsequently evaluated utilizing an Agilent 2100 Bioanalyzer (Agilent Technologies). RNA samples were pretreated to eliminate specific modifications that might impede the process (deacylation of 3′−aminoacyl RNA, removal of 3′−cP (2′,3′−cyclic phosphate), phosphorylation of 5′−OH, and demethylation of m1A and m3C). Library preparation was executed using the NEBNext® Multiplex Small RNA Library Prep Set for Illumina® (New England Biolabs). The procedural workflow incorporated 3′−adapter ligation, 5′−adapter ligation, cDNA synthesis, PCR amplification, and size selection of PCR products ranging from 134 to 160 bp. The quality and quantity of the libraries were assessed using the Agilent 2100 Bioanalyzer, followed by pooling equal amounts of each library. The pooled libraries were subsequently denatured with 0.1 M NaOH, diluted to a final concentration of 1.8 pM, and sequenced on an Illumina platform in accordance with the manufacturer's protocols.

## tRF-seq data analysis

Read trimming and clipping were done with cutadapt[58]. The resulting trimmed reads were consolidated into FASTA format. Read filtering, read mapping, nucleotide-wise coverage calculations, and genome feature-wise read quantification were performed using READemption2 version2.0.4[59] and the short-read mapper Segemehl version 0.3.4[60], applying an accuracy threshold of 95%. The references employed included the *E. coli* K-12 substr. DH10B genome (accession number NC_010473.1) and the T6 phage that was resequenced in the present study. The uniquely aligned reads were utilized to conduct the coverage calculations. The first 5′ bases were visualized using Integrative Genome Viewer version 2.16.0[55]. The ratio of the coverage, with respect to the presence and absence of Septu Ec[B88,] was mapped onto the tRNA secondary structure generated by VARNA[61].

## Cas13a-enabled phage genome editing

Phage genome editing was performed as previously described with minor modifications[62]. *E. coli* DH10B with the homologous recombination vector was cultured overnight in LB media at 200 rpm and 37 °C, then diluted 100-fold into fresh LB supplemented with 100 μg/ml kanamycin and 1 mM $CaCl_2$, resulting in ~8 × 10[6] CFU. Phages were added to reach an MOI of 0.01 (~8 × 10[4] PFU) and incubated for 3 h at 200 rpm and 37 °C. Cultures were centrifuged at 10,000 × $g$ for 2 min, and the supernatant filtered through a 0.45 μm filter. *E. coli* DH10B containing Cas13a vector was grown overnight in LB media at 200 rpm and 37 °C, then diluted 100-fold into fresh LB with 20 μg/ml chloramphenicol, 1 mM $CaCl_2$, and 10 nM aTc, reaching ~8 × 10[6] CFU. Lysates from the above DH10B with the homologous recombination vector were added for an MOI of 0.01 (~8 × 10[4] PFU) and incubated for 5 h at 200 rpm and 37 °C. Cultures were centrifuged, and supernatant filtered as mentioned above. To confirm mutations in phages, a bacterial lawn was prepared by mixing 300 μl of the overnight culture with Cas13a vector with 10 ml LB soft agar and 10 nM aTc, layered onto 1.5% LB agar with 20 μg/ml chloramphenicol. Ten-fold serial dilutions of potential edited and wild-type phages were prepared; 2 μl of each was spotted onto bacterial lawns and incubated at 37 °C overnight. The next day, single plaques from potential edited phages were isolated and resuspended in SM buffer. The infection of DH10B with the Cas13a vector was repeated to ensure the elimination of wild-type phages.

Finally, four independent plaques were isolated and stored at 4 °C until use. The edited phages were confirmed using plaque PCR and Cas13a selection.

## Experimental evolution via serial passaging

Fresh *E. coli* cultures were grown in LB medium supplemented with 1 mM $CaCl_2$ and infected with phage lysates derived from either wild-type T6 or the Δ*segB* mutant at a low MOI ~ 0.1. Cultures were incubated overnight at 37 °C, after which the supernatant was harvested and used to infect fresh *E. coli* cultures at an MOI ~ 0.1 in a total of five successive infection cycles. In each cycle, crude lysates were directly used as PCR templates without DNA purification. Segmentally amplified loci were PCR-amplified using KOD One polymerase (TOYOBO, KMM-101) following the manufacturer's protocol.

## SegB protein purification

The T7-inducible *segB* with a C-terminal twin-strepII tag expression plasmid was transformed into *E. coli* BL21(DE3). Overnight culture grown at 37 °C in LB medium supplemented with 0.5% glucose and 50 µg/ml kanamycin was 100-fold diluted into fresh LB medium supplemented with 0.5% glucose and 50 µg/ml kanamycin, and grown to $OD_{600} = 1.0$ at 37 °C. The preculture was 100-fold diluted into Overnight Express™ Instant TB Medium (Sigma-Aldrich) with 50 µg/ml kanamycin. The temperature was reduced to 18 °C at $OD_{600} = 0.6$, and cells were grown overnight. Cells were harvested and suspended in the lysis buffer (50 mM Tris-HCl, 300 mM NaCl, 5 mM $MgCl_2$, 1 mM DTT, pH 8.0) supplemented with 1 mM PMSF. Cells were lysed using a QSonica Q500 equipped with a microchip featuring a 1/4-inch irradiation surface diameter, with 15 s of sonication followed by 30 s on ice, repeated for 10 min at 25% amplitude. The lysate was cleared by centrifugation at 11,000 × *g* for 30 min. The cleared lysate was filtered and bound to Strep-Tactin Superflow Plus resin (QIAGEN), followed by washing with 10 column-volume lysis buffer. The proteins were eluted with lysis buffer supplemented with 2.5 mM desthiobiotin. Fractions containing SegB were concentrated and rebuffered with storage buffer (20 mM HEPES, 300 mM NaCl, 2.5 mM $MgCl_2$, 10% glycerol) in an Amicon Ultra-15 centrifugal concentrator (10,000 MWCO; Millipore).

## in vitro SegB cleavage assay

**PCR fragment.** The SegB endonuclease assay was performed as previously described with minor modifications[26]. Reactions were carried out in a total volume of 20 µl with 1× NEBuffer 2 [50 mM NaCl, 10 mM Tris-HCl, 10 mM $MgCl_2$, 1 mM DTT, pH 7.9] (NEB, #B7002S) in the presence or absence of 2 mM $MnCl_2$. Each reaction contained approximately 500 ng of PCR-amplified DNA fragments and an appropriate amount of SegB endonuclease. The reaction mixture was incubated at 30 °C for 1 h. The reaction was terminated by the addition of EDTA to a final concentration of 50 mM. Immediately, 0.5 µl of Proteinase K (Takara Bio) was added, and the mixture was further incubated at 55 °C for 30 min to ensure complete degradation of proteins. Reaction products were separated using 1% agarose gel in 1×TAE and 5% midori green extra (Nippon Genetics). The gels were then visualized under a Blue-Green LED light at a wavelength of 500 nm. Imaging of the gels was conducted using FAS-Digi PRO (Nippon Genetics).

**Synthetic dsDNA substrate.** dsDNA substrates were generated by annealing a 5′ FAM-labeled oligonucleotide with its unlabeled complementary strand in annealing buffer (10 mM Tris-HCl, pH 7.5, 50 mM NaCl). Oligonucleotides were heated to 95 °C for 5 min, and slowly cooled to room temperature to allow duplex formation. Reactions were carried out and terminated as described above. Each reaction contained 50 nM dsDNA substrate and 50 ng of purified nuclease protein. Reaction products were mixed with an equal volume of 2× Novex TBE-Urea sample buffer (Invitrogen, #LC6876), heated at 95 °C

for 3 min, and snap-cooled on ice prior to electrophoresis. Cleavage products were resolved by denaturing urea-PAGE Gels (20% polyacrylamide, 7 M urea). Fluorescently labeled DNA fragments were visualized using a LuminoGraph III (ATTO) with a Cy2 filter set.

## SITE-seq library preparation

All oligonucleotides used for the SITE-seq library preparation are listed in Supplementary Data 7. HMW gDNA was isolated from PEG-precipitated T6 phage lysates, scaled to 1 µg gDNA per SITE-seq sample. Purified gDNA was incubated with 50 ng purified SegB or mock control in a 20 µL reaction containing 1×NEBuffer 2 and 2 mM $MnCl_2$. Reactions were incubated at 30 °C for 1 h. Reactions were stopped by adding 1 µL of 1 M EDTA and 0.5 µL of Proteinase K (Takara), then incubated at 55 °C for 30 min. Equal volumes of SPRISelect beads (Beckman Coulter, #B23317) were added to the reaction and incubated at room temperature for 5 min. Beads were pelleted on a magnetic stand, washed twice with 85% ethanol, air-dried, and eluted in Ultra-Pure water. Thereafter, end repair and dA-tailing were performed using the NEBNext Ultra II End Repair/dA-Tailing Module (NEB, #E7546S) according to the manufacturer's specifications. Adapter 1 oligos were annealed in 1× annealing buffer (10 mM Tris-HCl pH 7.5, 50 mM NaCl, 1 mM EDTA) by heating to 95 °C for 5 min and cooling to room temperature. Ligation reactions containing end-prepped DNA, annealed Adapter 1, T4 DNA ligase buffer (NEB, #B0202S), and NEB Quick Ligase (NEB, #M2200S) were incubated at 20 °C for 30 min and then at 16 °C overnight. Ligation products were purified using 0.5× volumes of SPRISelect beads as described above. Adapter-ligated DNA was fragmented using the NEBNext dsDNA Fragmentase (NEB, #M0348S) at 37 °C for 1 h. Reactions were stopped with 0.5 M EDTA, diluted with water, and immediately subjected to SPRISelect purification. DNA was purified using 0.9× volumes of SPRISelect beads as described above. A second round of end repair/dA-tailing was performed using NEBNext Ultra II. Adapter 2 oligos were annealed in 1× annealing buffer. Ligation was carried out using the NEBNext Ultra II Ligation Module (NEB, #7595S) incubated at 20 °C for 30 min followed by 16 °C overnight. Dynabeads (Invitrogen, #11205D) were washed twice in 1× Block and Wash (B&W) buffer (prepared as 2× B&W: 10 mM Tris-HCl pH 7.5, 2 M NaCl, 1 mM EDTA) and resuspended in 2× B&W. Adapter-ligated DNA was added, and the mixture was incubated at room temperature with gentle inversion for 30 min. Beads were washed twice with 1× B&W buffer and once with 10 mM Tris-HCl (pH 8.5) before being resuspended in 10 mM Tris-HCl (pH 8.5). Bead-bound DNA was amplified using Q5 Hot-Start Master Mix (NEB, #M0494S) and Recovery PCR primers in 50 µL reactions. Following thermal cycling, supernatants were recovered after magnetic separation. A portion of the PCR product was diluted and used as template for indexing PCR with NEBNext Multiplex Oligos for Illumina (NEB, #E7600S) and Q5 Master Mix. Indexed libraries were purified using 0.7× SPRISelect beads, washed, dried, and eluted in UltraPure water to obtain final 50 µL libraries. Libraries were sequenced on an Illumina NextSeq 2000 platform using paired-end 150-bp reads.

## Peak calling of SegB cleavage sites

Paired-end SITE-seq reads were aligned to the T6 reference genome using bwa mem with sample-specific read group information. SAM output was converted to BAM, name-sorted, and processed with samtools fixmate, coordinate sorting, and samtools markdup to generate duplicate-marked BAM files[57]. Basic alignment statistics were obtained with samtools flagstat and samtools idxstats[57]. For 5′ end analysis, only first mates (R1) were retained. R1 reads were extracted from the duplicate-marked BAM files, coordinate-sorted, and indexed, yielding per-sample R1 BAMs for downstream peak calling. Differential 5′ end peaks between SegB-treated and mock-treated samples were identified using a custom Python workflow that directly parses R1 BAM files with pysam. For each replicate, the genomic position of the R1 5′

end was determined (forward: reference_start; reverse: reference_end −1), and reads failing basic mapping filters or with short aligned length (<40 nt) were excluded. This yielded a per-base 5′ end count array for each replicate. Counts were converted to CPM per replicate and summed within each condition (SegB or mock). The aggregated profiles were smoothed using a 5-bp moving average, and $\log_2$ fold change was computed using a small pseudocount.

Candidate summits were defined as local maxima in the smoothed SegB profile that satisfied multiple stringency criteria: a minimum SegB signal of 1.0 CPM, a minimum $\log_2 FC$ of 2, and a minimum SegB-to-mock ratio of 1.2 at the summit position. To ensure biological reproducibility, replicate consistency was enforced at the raw-count level: at least two SegB replicates were required to show a maximum raw 5′ end count of ≥5 within a ±1 bp window centered on the candidate summit. Summits located within 3 bp of each other were merged, and within each merged cluster the position with the highest smoothed SegB signal was selected as the final summit. Final peaks were reported as 3-bp genomic intervals centered on the summit and output in BED format.

## MEME motif analysis

To identify sequence motifs associated with SegB-dependent 5′ end peaks, we extracted genomic sequences surrounding peak summits. A ±20 bp window around each summit was retrieved from the T6J reference genome using pysam, ignoring peaks whose extended window exceeded contig boundaries. De novo motif discovery was performed using MEME in ZOOPS mode, searching for up to five motifs[63]. Analyses were run on DNA mode with reverse-complement scanning enabled, using a motif width range of 5–15 bp.

## Mass photometry analysis

The Refeyn TwoMP mass photometer was utilized for conducting mass photometry experiments. Mass measurements were performed in a 1× phosphate buffer saline (PBS) solution. Calibration was performed using an equimolar mixture of bovine serum albumin (67 kDa) and human IgG (150 kDa), diluted in 1×PBS. Sample stocks were added into wells containing the buffer to reach the nanomolar concentrations necessary for mass photometry. Data acquisition was executed using AcquireMP, while data analysis was carried out with DiscoverMP. Three measurements were taken in each buffer. Masses were determined using the appropriate calibration. The calculated masses for the three measurements are as follows: $46 \pm 8$ kDa, $43 \pm 16.8$ kDa, and $43 \pm 8.4$ kDa.

## Phylogenetic analysis

Homologs of the T6 phage SegB sequence were identified via BLASTp (E-value cutoff: 1e-5; minimum identity: 30%; coverage threshold: 80%), yielding 79 homologs. The sequences were clustered using MMseqs2 easy-cluster with 95% identity and 95% coverage thresholds[64]. Subsequently, the clustered sequences were aligned employing MAFFT (v7.51136) with default settings[65], and an HMM profile was built utilizing hmmbuild. Protein sequences from bacteriophages, 2,091,036 in total, and archaeal viruses, 18,046 in total, were acquired from GenBank. Homologs were extracted through hmmsearch with a GA cut threshold of 20, resulting in the identification of 1832 homologs. Among these, 386 sequences that met the criteria of profile identity not exceeding 95% were selected for phylogenetic analysis. A phylogenetic tree was constructed employing the best-fitting model (PMB + F + R7) using IQ-TREE (version 1.6.12) with the following command parameters: -bb 1000 -m TESTNEW -nt AUTO -wbtl -pre[66]. The online tool iTOL v6 facilitated the visualization of the phylogenetic tree[67]. The phylogenetic tree for Gp49.2 was also constructed, as mentioned above, using the best-fitting model (VT + G4) in IQ-TREE.

## Gene neighborhood analysis

Sequences ranging from 1000 bp upstream to 1000 bp downstream of each segB homolog were retrieved as a FASTA file utilizing a custom Python script. Bacterial tRNAs were searched for the sequences using tRNAscan-SE version 2.0.12 with default settings[28]. To assess the confidence interval of the median difference between segB-adjacent tRNAs and tRNAs throughout the genome, we applied nonparametric bootstrap resampling (10,000 iterations). In each iteration, random samples with replacement were drawn independently from each group, and the difference in medians was computed. A 95% confidence interval was determined by calculating the 2.5th and 97.5th percentiles of the resulting distribution of median differences. To assess the density of tRNAs (the number of tRNA genes per base pair) between segB-adjacent sequences and the genome-wide background, we used the Wilcoxon signed-rank test, as the data were paired and not normally distributed. Only genomes for which both the SegB locus and genome-wide tRNA counts were available were included in the analysis.

## Data visualization and figure preparation

The proteome composition distance (PCD) matrix of the BASEL collection phages was calculated using LoVis4u[68]. Hierarchical clustering of the PCD matrix, using the average linkage method, was performed to define the phage order and dendrogram. The heatmap of the defense profile was generated using the R package ComplexHeatmap[69]. Aligned amino acid sequences of representative PtuAs and PtuBs were visualized using JalView[70]. GraphPad Prism 10.1.1 was used to visualize bar graph showing the PFU and bacterial growth. All the figures with this manuscript were generated with Adobe Illustrator 2024.

## Statistics and reproducibility

No statistical method was used to predetermine sample size. Sample sizes were chosen based on standard practice in the field and the availability of biological material. No data were excluded from the analyses. The experiments were not randomized. The investigators were not blinded to allocation during experiments and outcome assessment. Statistical analyses were performed using Python 3. For the analysis shown in Fig. 4c, a one-sided paired Wilcoxon signed-rank test was used to compare tRNA gene density between segB-proximal regions and genome-wide regions across phage genomes. Details of replicate numbers and experimental reproducibility are provided in the corresponding figure legends.

## Reporting summary

Further information on research design is available in the Nature Portfolio Reporting Summary linked to this article.

## Data availability

The tRNA fragment sequencing raw and processed data are available on the Gene Expression Omnibus (GEO) repository under accession number GSE303005. SITE-seq raw and processed data are available on the GEO repository under accession number GSE315621. Raw sequencing data for escaper phages are available in the DDBJ Sequenced Read Archive under the accession number DRA021756. Source data are provided with this paper.

## Code availability

Custom scripts used for data analysis are available on Zenodo under https://doi.org/10.5281/zenodo.18158113 (https://doi.org/10.5281/zenodo.18158113).

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

## Acknowledgements

We would like to thank Alexander Harms for providing the BASEL phage collection. We thank Dr. Chika Arai, Dr. Yo Sugawara and Dr. Motoyuki Sugai for providing *Escherichia coli* strains from the Japanese Anti-microbial Resistant Bacterial Bank (JARBB). We would like to thank Matthew Imanaka for extracting and sequencing genomic DNA from *E. coli* JBBDAEE-19-0007. We thank other Kiga lab members in the Department for Drug Development at NIID for their critical comments on the project. This work was supported by the Japan Agency for Medical Research and Development (Grant Nos. JP24gm1610002, JP23wm0325065, JP24fk0108698, and JP25ym0126806 to K.K.; JP22fk0108562 and JP23fk0108599 to K.C.), JSPS KAKENHI (Grant Nos. 21H02110 and 25K21732 to K.K.; 23K19475 and 25K18809 to K.C.), and Shionogi infectious disease research promotion grant and G-7 scholarship grant to K.K. V.H. was supported by the Knut and Alice Wallenberg Foundation project grant 2020-0037 to V.H., Vetenskapsrådet grants 2021-01146 and ÄR-MH 2024-06059 as well as Göran Gustafsson Foundation for Research in Natural Sciences and Medicine (the Göran Gustafsson Prize to V.H.). The funders had no role in the study design, data collection and analysis, decision to publish, or preparation of the manuscript.

## Author contributions

K.C. and K.K. conceived, designed, and interpreted the experiments and wrote the manuscript. K.C. conducted the experiments and generated all figures. K.C. and A.H.A. constructed plasmids. K.C. and A.A.E. performed computational analyses. K.C., A.H.A., and I.T. screened T6 phage infection across *E. coli* strains expressing defense systems. M.H. and K.H. sequenced the SITE-seq library. K.W., V.H., and K.K. contributed critical resources and advice. All authors contributed to the article.

## Competing interests

The authors declare no competing interests.
