## [Transparent Peer Review file · Nature Communications]

Phage homing endonuclease amplifies anti-defense genes to evade bacterial immunity

Corresponding Author: Dr Kotaro Kiga

Version 0:

Reviewer comments:

Reviewer #1

(Remarks to the Author)

This manuscript investigates an intriguing mechanism of phage counterdefense, detailing how the T6 phage escapes the Septu bacterial immune system. The authors show that the Septu defense functions by cleaving a specific phage tRNA (tRNATyr). In response, the phage utilizes the homing endonuclease SegB to facilitate a segmental amplification of the genomic region containing the tRNATyr gene. This amplification leads to increased tRNATyr expression, which is proposed to overcome the defense's tRNA cleavage activity. The authors also demonstrate that SegB can mediate this amplification for other anti-defense genes, suggesting this is a general and flexible strategy for phages to rapidly adapt and evade diverse bacterial defenses.

The experimental work presented is robust and compelling. My primary concerns relate to the manuscript's accessibility, a missed opportunity for broader contextualization, and remaining questions on the mechanism of SegB.

A limitation of the paper is its accessibility. The writing is often overly concise, seemingly to fit a restrictive word count that may not be necessary. This conciseness, while sometimes beneficial, here obscures the impact of the work for a broader audience.

1. The introduction, in particular, is written for a narrow field of specialists (only 2 brief paragraphs). It assumes a significant amount of prior knowledge, which may alienate readers who are not already familiar with the specific defense systems and phage-host interactions being discussed. A slightly broader setup would make the work more approachable and highlight its general implications more effectively.

2. There are several instances throughout the results where data is presented without adequate explanation. Figures are included, but their full significance is not always unpacked in the text. This leaves the reader to do the work of interpreting the data, rather than being guided by the authors' narrative, and decreases the accessibility.

For example:

Lines 80-82 refer to Extended Data Fig. 3a-b, which appears to be data ruling out observed SNPs. It seems the authors may have hypothesized that gp234 was a trigger for the system but found no evidence for this in toxicity assays. However, none of this is explained in the text. This is a clear case where the data should either be fully explained or omitted to avoid confusion.

3. I have several questions regarding the SegB mechanism:

a. Fig 3B: The substrate for the gel is not defined. It would be clearer to crop the gel (removing the top) and include a schematic of the substrate.

b. SegB Self-Targeting: A major unresolved question is how the phage survives the nuclease activity of its own enzyme. Does SegB cut the unamplified, wild-type region? How does the phage avoid self-destruction? Typically, homing destroys the cut site.

c. Nuclease Activity: Is the in vivo activity of SegB dependent on its nuclease function? This could be tested by creating a catalytically-dead mutant, which should phenocopy the deltaSegB mutant. This would also serve as a much-needed control for the in vitro cutting assays, given the protein purification.

D. Cleavage Site: With the in vitro cleavage assays, can the authors identify the specific recognition site? This would be a powerful next step, allowing them to engineer the site near other counter-defense genes to test the model's generality.

Minor points:

1. The authors compellingly demonstrate that the observed genomic amplification is unstable and collapses in the absence of selective pressure. This finding is highly reminiscent of the "accordion model" of genome dynamics observed in poxviruses. This parallel is striking, and the manuscript would be significantly strengthened by a discussion of this, placing

their findings within a broader, established context of rapid adaptation via reversible gene amplification. As written, this feels like a missed opportunity to connect their work to a more general biological principle.

2. Line 37: The phrasing "Septu defense FROM the E.coli clinical strain" is potentially misleading, as the experiments were not conducted in the clinical strain itself. This should be clarified.
3. Line 53 & Fig 1B: There is a contradiction. The text states T6 has intermediate defense and T4 shows a significant reduction, but the Fig 1B heatmap shows T2 as the most restricted (darkest green), not T4.
4. Fig 2B: The lawns for the SegB expression in E. coli appear very thin. This toxicity makes it uninterpretable, and this panel should be excluded.
5. Fig 2h: It is unclear how a tRNA cleavage site can be mapped when an identical pattern appears in the control lane without Septu.
6. Ext. Data Fig 2B: The PFU (final/initial) ratio decreases as MOI increases, even in the Septu(-) host. This is a confusing result, possibly indicating lysis from without, which calls the experimental setup into question.
7. Ext. Data Fig 4: The data implies T4 cannot escape Septu because it lacks the correct tRNA^{Tyr} gene, even though it possesses SegB. If this is the authors' hypothesis, it should be explicitly stated.
8. Ext. Data Fig 9c: In Ext. Data Fig 9c, T4 deltaSegB seems to escape ToxIN (based on spot assay), which should not happen if SegB is required for amplification. This is not reflected in the PCR. Is there another mode of escape here? The legends for "repeats" are also unclear.
9. Ext. Data Fig 8a: The "cycles" aspect of this figure is not clearly explained.
10. Lines 160-163: The phrasing here suggests the finding is novel to this paper, when it appears to be a model drawn from existing literature. This should be rephrased.
11. Lines 179-181: The idea that SegB moves "sporadically between phage genomes" is misleading. It may be that our view of phage host range is too limited; phages might co-infect bacteria more often than they productively infect, allowing for more frequent gene transfer.
12. Line 183: The claim that SegB is more often in Myoviruses is likely a significant database bias
13. Line 244: The statement "broader strategy employed by phages" is an overstatement. T4 and T6 are highly similar (e.g., The T-even type genomes share 85-95% ORF homology with one another and >90% nucleotide sequence identity between most of their shared alleles). This high degree of relatedness, which may not be obvious to all readers, should be acknowledged, and the claim of a "broader" strategy should be toned down.
14. Line 249: To appreciate the in trans effect, they should state the genomic distance (in Kb) between SegB and the loci it is amplifying.

Reviewer #2

(Remarks to the Author)

The manuscript by Chihara et al describes the impact of phage-encoded homing endonuclease SegB on the outcome of bacteria-phage interactions. SegB appears to initiate cleavage and subsequent homologous recombination to allow expansion of phage genomic loci that result in increased resistance against host defences. This was determined using Septu as a model, but then recapitulated for OLD and ToxIN. In the first case, tRNA-Tyr is expanded. For OLD, it appears to be a new inhibitor that binds OLD, and then for ToxIN, it is expansion of known inhibitor TifA. This indicates SegB and expansion caused by SegB could be a common route leading to resistance in phages against bacterial defences.

The study will no doubt be of wide interest. Overall I find the story compelling and generally well supported. I think there are some details that need tidying up:

Line 35 - Intro is very short. Please provide paragraph on homing endonucleases to help non-experts and remind "experts".

Line 48 - what is the copy number, is this induced or using native promoter?

Line 52 - I am confused about the colouring of T2/T4/T6 defense in 1B and then EOPs in 1C. They do not seem to match. 1B looks like strong defense vs T2, but in 1C there is little impact. Please clarify.

Line 89 - In 1G, why would primer pair 1 give a ladder in the escapes? If the population has been isolated to plaque purity, then there would be a single expansion providing a single larger amplicon. The ladder implies that each phage sample is a mixed population still.

Line 104 - I don't understand this sentence - the only expanded genes are tRNA-Tyr and truncated SegB. Why are the other tRNAs a consideration?

Line 114 - You suggest phage tRNA reduces degradation of host tRNA, but in 2C you see greater degradation of host (at least to my eyes). Please clarify as this claim seems unsupported.

Fig 3 - Use kb not kbp

Extended figs - use prime symbols, not apostrophes

Fig 3b - very little info provided. Why is there so little impact of adding the metal, it appears already active. In extended fig 5b also provide AFold scores

Line 152 - state you did see growth inhibition with WT

Line 209 - ladders again, please clarify

Line 233 - a pull-down showing interaction of 49.2 with OLD would be far more compelling. However, consider this only a suggestion rather than a requirement. Maybe a better route would be to KO ket surface interaction residues of 49.2 and show no longer protects?

Version 1:

Reviewer comments:

Reviewer #1

(Remarks to the Author)

The authors have greatly improved the manuscript in response to our suggestions. Below are a few minor points that remain:
line 57 from an E.coli clinical strain (not the)

when the repeats collapse without immune pressure, is the SegB truncation is restored to the WT copy? please add this information when discussing the collapse.

Line 190 sounds as if nuclease activity is necessary for viral propagation, it should be specified that it is indispensable for segmental amplification of the locus of interest in vivo.

Line 217 - reproducible cleavage sites implies replicates of the experiment were performed, please include this information in the legend

Line 227 - they claim that motif-disrupted substrates showed little to no cleavage in Fig 3h; while the 30nt mut substrates appears more abundant, cleavage products are still present and appear to my eye equal to the Conc. motif. If claiming cleavage efficiency, quantification of replicate assays should be provided.

Line 327: misleading; Gp49.2M13A still functions for T4.

(Remarks on code availability)

Reviewer #2

(Remarks to the Author)

The authors have done a thorough job of addressing my concerns. I congratulate them on a lovely story. No further suggestions from me.

(Remarks on code availability)

General responses

1. To address the reviewer's comments and to incorporate the additional experimental data generated during revision, we reorganized the original Fig. 3 into two separate figures: a new Fig. 3 focusing on the mechanistic aspects of SegB-mediated amplification and a new Fig. 4 presenting the phylogenetic analysis. This restructuring was performed to improve clarity and to more logically integrate the newly acquired data.
2. During the revision process, new experiments were conducted, and Dr. Masanori Hashino and Dr. Kazuhiro Horiba made substantial contributions to these additional data. Therefore, we have added them as a co-author. The author list has been updated, and all authors have approved this modification.
3. In response to the reviewers' comments, we re-evaluated the terminology used to describe the phage infection experiments. We recognized that the term *repeated infection assay* could potentially be misinterpreted as referring to experimental replication rather than sequential rounds of phage infection, lysis, and lysate transfer. To avoid this ambiguity and to more accurately reflect the experimental design, we have replaced this term throughout the manuscript with *experimental evolution via serial passaging* or simply *serial passaging experiment*. This revision clarifies that the experiments consist of consecutive passages in which fresh *E. coli* cultures are infected with lysates from the previous cycle, rather than technical or biological replicates. All relevant sections and figure legends have been updated accordingly.

Reviewer #1 (Remarks to the Author):

This manuscript investigates an intriguing mechanism of phage counterdefense, detailing how the T6 phage escapes the Septu bacterial immune system. The authors show that the Septu defense functions by cleaving a specific phage tRNA (tRNATyr). In response, the phage utilizes the homing endonuclease SegB to facilitate a segmental amplification of the genomic region containing the tRNATyr gene. This amplification leads to increased tRNATyr expression, which is proposed to overcome the defense's tRNA cleavage activity. The authors also demonstrate that SegB can mediate this amplification for other anti-defense genes, suggesting this is a general and flexible strategy for phages to rapidly adapt and evade diverse bacterial defenses.

The experimental work presented is robust and compelling. My primary concerns relate to the manuscript's accessibility, a missed opportunity for broader contextualization, and remaining questions on the mechanism of SegB.

A limitation of the paper is its accessibility. The writing is often overly concise, seemingly to fit a restrictive word count that may not be necessary. This conciseness, while sometimes beneficial, here obscures the impact of the work for a broader audience.

We thank the reviewer for the positive evaluation of our work and for highlighting these important overarching concerns. We have addressed these issues in the revised Introduction, Results, and Discussion, as detailed in our responses below.

1. The introduction, in particular, is written for a narrow field of specialists (only 2 brief paragraphs). It assumes a significant amount of prior knowledge, which may alienate readers who are not already familiar with the specific defense systems and phage-host interactions being discussed. A slightly broader setup would make the work more approachable and highlight its general implications more effectively.

Thank you for this helpful suggestion. In the revised manuscript, we have expanded the Introduction to provide a broader conceptual framework. Specifically, we now include additional background on homing endonuclease genes (HEGs), their mechanisms of mobility, and their roles in phage genome diversification and competition (lines 44–54). These additions offer non-specialist readers a clearer understanding of the evolutionary context that motivates our study and more effectively highlight the broader implications of SegB-mediated gene amplification.

2. There are several instances throughout the results where data is presented without adequate explanation. Figures are included, but their full significance is not always unpacked in the text. This leaves the reader to do the work of interpreting the data, rather than being guided by the authors' narrative, and decreases the accessibility.

We thank the reviewer for pointing out that several parts of the Results section lacked sufficient explanation. In response, we have carefully revised the manuscript to improve the clarity and accessibility of the narrative. Specifically, we expanded descriptions of the experimental logic and the interpretation of data at multiple points in the Results, including the experimental evolution of phages (lines 177–182), the in vitro cleavage assays (lines 191–197), the selection of additional tRNA-targeting defense systems (lines 277–281), and the phylogenetic analysis workflow (lines 248–251). We also expanded the Introduction and Discussion to provide broader biological context for readers outside the immediate field.

For example:

Lines 80-82 refer to Extended Data Fig. 3a-b, which appears to be data ruling out observed SNPs. It seems the authors may have hypothesized that gp234 was a trigger for the system but found no evidence for this in toxicity assays. However, none of this is explained in the text.

This is a clear case where the data should either be fully explained or omitted to avoid confusion.

We thank the reviewer for noting the lack of explanation for Supplementary Fig. 3a–b. We have now revised the text to clearly state that (i) escape phages share no common SNPs or indels, and (ii) although two isolates carry a mutation in gp234, co-expression of the mutated tail-sheath gene with Septu Ec^{B88} does not trigger toxicity. This clarification is now included in the revised Results, lines 103–110.

3. I have several questions regarding the SegB mechanism:

a. Fig 3B: The substrate for the gel is not defined. It would be clearer to crop the gel (removing the top) and include a schematic of the substrate.

We thank the reviewer for this suggestion. In the revised Fig. 3b, we have cropped the gel to remove the wells and added a schematic of the DNA substrate directly above the gel.

b. SegB Self-Targeting: A major unresolved question is how the phage survives the nuclease activity of its own enzyme. Does SegB cut the unamplified, wild-type region? How does the phage avoid self-destruction? Typically, homing destroys the cut site.

We thank the reviewer for highlighting this important point. To address it, we added a concise paragraph in the Discussion (lines 386–403) explaining how SegB may avoid lethal self-targeting. Specifically, we note that (i) T-even phages possess a robust recombination-dependent repair system, (ii) SITE-seq shows that SegB cleavage is limited and sequence-dependent, (iii) GIY-YIG endonucleases generally cut recipient but not donor alleles, and (iv) phage DNA-binding proteins such as MotB may restrict access to potential cleavage sites *in vivo*. This clarification is now included in the revised manuscript.

c. Nuclease Activity: Is the *in vivo* activity of SegB dependent on its nuclease function? This could be tested by creating a catalytically-dead mutant, which should phenocopy the deltaSegB mutant. This would also serve as a much-needed control for the *in vitro* cutting assays, given the protein purification.

We thank the reviewer for this important suggestion. To assess whether SegB's *in vivo* activity depends on its nuclease function, we generated catalytically inactive SegB mutants (Y17A/G19A and Y17F/G19A). As an initial functional assessment, we examined the effects of overexpressing these variants *in vivo*. In contrast to wild-type SegB, neither mutant caused detectable growth inhibition or toxicity, consistent with a loss of SegB activity under overexpression conditions (revised Supplementary Fig. 5d–e).

We also purified the SegB mutant proteins and found that both variants lacked detectable DNA cleavage activity *in vitro*, providing a stringent negative control for the *in vitro* cleavage assays (revised Supplementary Fig. 5c). Finally, a T6 phage strain encoding SegB^{Y17A/G19A} failed to support segmental amplification of the locus in repeated infection assays, phenocopying the $\Delta segB$ mutant (revised Fig. 3a, right). Due to technical limitations of the CRISPR–Cas13a–based mutant enrichment approach, *in vivo* experiments could not be performed with the SegB^{Y17F/G19A} variant.

Together, these results indicate that the nuclease activity of SegB is required for its function *in vivo*. The corresponding data and text have been added to the revised manuscript (Results, lines 186–205).

D. Cleavage Site: With the *in vitro* cleavage assays, can the authors identify the specific recognition site? This would be a powerful next step, allowing them to engineer the site near other counter-defense genes to test the model's generality.

We thank the reviewer for this valuable suggestion. To identify a potential SegB recognition site, we reasoned that it would be necessary to map SegB cleavage sites genome-wide. We therefore performed a genome-wide SITE-seq assay using purified SegB and high-molecular-weight phage DNA. SITE-seq was originally developed to assess genome-wide cleavage specificity of CRISPR–Cas9 nucleases (PMID: 28459459) and is well suited for detecting sequence-dependent DNA cleavage *in vitro*.

Using a cutoff of $\text{Log}_2(\text{SegB}/\text{Mock}) > 2$, we identified 83 SegB-dependent cleavage sites distributed across the phage genome (revised Fig. 3d–f). Motif discovery analysis (MEME) using ± 20 bp sequences surrounding these sites revealed a conserved DNA motif, suggesting sequence-specific recognition by SegB (revised Fig. 3g).

To validate this motif experimentally, we performed *in vitro* cleavage assays using synthetic dsDNA substrates. SegB efficiently cleaved substrates containing the identified motif, whereas substrates lacking the motif showed minimal cleavage under identical conditions (revised Fig. 3h). Together, these data suggest that SegB functions as a sequence-dependent nuclease and provide a mechanistic basis for the locus-specific genomic amplification observed *in vivo*. The corresponding results have been added to the revised manuscript (Results, lines 211–230).

Minor points:

1. The authors compellingly demonstrate that the observed genomic amplification is unstable and collapses in the absence of selective pressure. This finding is highly reminiscent of the "accordion model" of genome dynamics observed in poxviruses. This parallel is

striking, and the manuscript would be significantly strengthened by a discussion of this, placing their findings within a broader, established context of rapid adaptation via reversible gene amplification. As written, this feels like a missed opportunity to connect their work to a more general biological principle.

We appreciate this insightful comment highlighting the striking parallel between our findings and the “genomic accordion” model described for poxviruses. We fully agree that this connection places our results within a broader and well-established framework of rapid, reversible viral adaptation. In the revised Discussion (lines 404–413), we have added a new paragraph explicitly comparing SegB-mediated, reversible genomic amplifications in phage T6 with the genomic accordion dynamics described in poxviruses. We discuss how, in both systems, gene copy number can expand rapidly under selective pressure and subsequently collapse once that pressure is relieved, emphasizing genome plasticity as a general evolutionary strategy rather than a virus-specific peculiarity. We believe that this expanded discussion strengthens the conceptual impact of the study by framing phage adaptive genome dynamics within a unifying principle shared across diverse large DNA viruses.

2. Line 37: The phrasing "Septu defense FROM the E.coli clinical strain" is potentially misleading, as the experiments were not conducted in the clinical strain itself. This should be clarified.

We thank the reviewer for noting this ambiguity. We have revised the wording from “in” to “from” to clarify that the Septu system was derived from the clinical *E. coli* strain but not tested in that strain.

3. Line 53 & Fig 1B: There is a contradiction. The text states T6 has intermediate defense and T4 shows a significant reduction, but the Fig 1B heatmap shows T2 as the most restricted (darkest green), not T4.

We thank the reviewer for catching this error. The heatmap in Fig. 1B was incorrect in the original submission; we have now corrected the figure so that the visual data accurately match the text.

4. Fig 2B: The lawns for the SegB expression in *E. coli* appear very thin. This toxicity makes it uninterpretable, and this panel should be excluded.

We agree with the reviewer that SegB expression causes strong toxicity in this assay, making Fig. 2B difficult to interpret. As suggested, we have removed this panel from the revised manuscript.

5. Fig 2h: It is unclear how a tRNA cleavage site can be mapped when an identical pattern appears in the control lane without Septu.

We thank the reviewer for raising this point and apologize that the logic connecting Figs. 2h and Fig. 2i was not sufficiently clear. Importantly, cleavage site assignment was not based on Fig. 2h in isolation, but on quantitative comparison of read coverage between +Septu and –Septu conditions. As shown in Fig. 2i, mapping the +Septu/–Septu coverage ratio onto the T6 tRNA^{Tyr} secondary structure revealed a single, prominent enrichment at cytosine 48, which we defined as the Septu-dependent cleavage site. The multiple peaks observed in Fig. 2h likely reflect the general instability of T6 tRNA^{Tyr}, as noted in the text, whereas only C48 shows consistent Septu-dependent enrichment. To clarify this distinction, we have revised the Results text to explicitly state that Fig. 2i provides the basis for cleavage site identification, while Fig. 2h shows overall cleavage patterns and background instability (lines 164–168).

6. Ext. Data Fig 2B: The PFU (final/initial) ratio decreases as MOI increases, even in the Septu(-) host. This is a confusing result, possibly indicating lysis from without, which calls the experimental setup into question.

We thank the reviewer for highlighting this point. The reduced PFU ratio of T4 at high MOI in the Septu(-) host is most likely due to lysis from without. This effect is independent of Septu activity and therefore does not affect our interpretation of T6-specific escape. To avoid confusion, we have added a brief clarification to the text, lines 87–89.

7. Ext. Data Fig 4: The data implies T4 cannot escape Septu because it lacks the correct tRNA^{Tyr} gene, even though it possesses SegB. If this is the authors' hypothesis, it should be explicitly stated.

Thank you for this insightful comment. We agree that the data in Supplementary Fig. 4 suggest that T4 fails to escape the Septu system because it lacks a functional tRNA^{Tyr} gene, despite encoding the homing endonuclease SegB. We have now revised the manuscript to explicitly state this interpretation in the Results, lines 150–152.

8. Ext. Data Fig 9c: In Ext. Data Fig 9c, T4 deltaSegB seems to escape ToxIN (based on spot assay), which should not happen if SegB is required for amplification. This is not reflected in the PCR. Is there another mode of escape here? The legends for "repeats" are also unclear.

Thank you for pointing this out. Although we did not further investigate this pathway in the present study, one plausible explanation is that mutations may have arisen in genes required

to trigger the ToxIN system, thereby preventing its activation. To address this, we have now added a brief clarification in the Results section, lines 341–344.

Second, we apologize for the ambiguous use of the term “repeats” in the figure legend. In this experiment, “repeats” referred to sequential rounds of phage infection, lysis, and lysate recovery in *E. coli*. To avoid confusion, we have replaced “repeats” with “infection cycles” throughout the figure legend and text, as well as figures.

9. Ext. Data Fig 8a: The “cycles” aspect of this figure is not clearly explained.

We thank the reviewer for this comment and apologize for the lack of clarity. In Supplementary Fig. 8a, “infection cycles” refer to sequential rounds of phage infection, lysis, and lysate recovery, in which fresh *E. coli* cultures were infected with supernatants from the previous cycle. The first through fifth infection cycles are shown. We have now clarified this terminology in the figure legend.

10. Lines 160-163: The phrasing here suggests the finding is novel to this paper, when it appears to be a model drawn from existing literature. This should be rephrased.

We thank the reviewer for pointing this out. We have revised the sentence to clarify that this description reflects a model supported by previous literature.

11. Lines 179-181: The idea that SegB moves “sporadically between phage genomes” is misleading. It may be that our view of phage host range is too limited; phages might co-infect bacteria more often than they productively infect, allowing for more frequent gene transfer.

Thank you for this insightful comment. To avoid implying an overly limited frequency, we have revised the wording to emphasize that SegB can move between phages infecting different host taxa, without speculating on the absolute frequency of such events.

12. Line 183: The claim that SegB is more often in Myoviruses is likely a significant database bias

We appreciate the reviewer’s point. We have revised the text to clarify that this observation could be influenced by database sampling bias and should not be overinterpreted (lines 259–261).

13. Line 244: The statement “broader strategy employed by phages” is an overstatement. T4 and T6 are highly similar (e.g., The T-even type genomes share 85-95% ORF homology with one another and >90% nucleotide sequence identity between most of their shared alleles).

This high degree of relatedness, which may not be obvious to all readers, should be acknowledged, and the claim of a "broader" strategy should be toned down.

We thank the reviewer for this important point. We have revised the text to acknowledge their close relatedness and to tone down the claim by stating that our findings suggest a shared strategy within T-even phages rather than a broadly conserved phage-wide mechanism (lines 344–347).

14. Line 249: To appreciate the in trans effect, they should state the genomic distance (in Kb) between SegB and the loci it is amplifying.

We thank the reviewer for this helpful suggestion. We have now added the genomic distances between *segB* and the loci it amplifies. In T6, *gp49.2* is located ~22 kb from *segB*, and in T4, *tifA* is located ~52 kb away. This information has been incorporated into the revised manuscript, lines 352–355.

Reviewer #2 (Remarks to the Author):

The manuscript by Chihara et al describes the impact of phage-encoded homing endonuclease SegB on the outcome of bacteria-phage interactions. SegB appears to initiate cleavage and subsequent homologous recombination to allow expansion of phage genomic loci that result in increased resistance against host defences. This was determined using Septu as a model, but then recapitulated for OLD and ToxIN. In the first case, tRNA-Tyr is expanded. For OLD, it appears to be a new inhibitor that binds OLD, and then for ToxIN, it is expansion of known inhibitor TifA. This indicates SegB and expansion caused by SegB could be a common route leading to resistance in phages against bacterial defences. The study will no doubt be of wide interest. Overall I find the story compelling and generally well supported. I think there are some details that need tidying up:

We thank the reviewer for their positive assessment of our work and for the constructive suggestions. We have addressed all specific points below.

Line 35 - Intro is very short. Please provide paragraph on homing endonucleases to help non-experts and remind "experts".

We thank the reviewer for this constructive suggestion. In the revised manuscript (lines 44–54), we have expanded the Introduction by adding a dedicated paragraph describing homing endonuclease genes (HEGs), their mechanism of mobility, their prevalence in bacteriophages, and their roles in phage genome diversification and competition.

Line 48 - what is the copy number, is this induced or using native promoter?

We thank the reviewer for the question. Septu Ec^{B88} was cloned on a p15A-origin plasmid (~10 copies per cell) and expressed from its native promoter. We have clarified this in the revised text, line 69.

Line 52 - I am confused about the colouring of T2/T4/T6 defense in 1B and then EOPs in 1C. They do not seem to match. 1B looks like strong defense vs T2, but in 1C there is little impact. Please clarify.

We thank the reviewer for pointing out this inconsistency. The heatmap in Fig. 1B contained an error in the original submission, which caused it to mismatch the EOP data in Fig. 1C. We have corrected Fig. 1B in the revised manuscript so that the defense levels now accurately reflect the EOP measurements.

Line 89 - In 1G, why would primer pair 1 give a ladder in the escapes? If the population has been isolated to plaque purity, then there would be a single expansion providing a single larger amplicon. The ladder implies that each phage sample is a mixed population still.

We thank the reviewer for this insightful question. We believe that the ladder observed with primer pair 1 does not reflect incomplete plaque purification, but rather the intrinsic heterogeneity of segmental amplification. Segmental amplification in T-even phages is likely transient and recombination-driven, allowing copy numbers to expand and contract during intracellular replication. Consistent with this view, our experimental evolution via serial passaging shows that amplified bands progressively diminish once immune pressure is removed, indicating that amplification is inherently unstable. Consequently, even plaque-purified escapers likely contain phages with different repeat numbers, producing a ladder of products rather than a single discrete band.

Line 104 - I don't understand this sentence - the only expanded genes are tRNA-Tyr and truncated SegB. Why are the other tRNAs a consideration?

We thank the reviewer for this clarification request. Our intention was not to imply that multiple tRNAs were expanded, but rather that the amplified locus contains several tRNA genes, any of which could have been candidates prior to determining that only tRNA-Tyr is functionally relevant. We have revised the sentence to make this distinction explicit (lines 133-135).

Line 114 - You suggest phage tRNA reduces degradation of host tRNA, but in 2C you see greater degradation of host (at least to my eyes). Please clarify as this claim seems unsupported.

We thank the reviewer for this clarification request and agree that the original wording could be misinterpreted. We did not intend to suggest that phage-derived tRNA^{Tyr} broadly prevents degradation of host tRNA during infection. Indeed, Fig. 2C shows substantial depletion of host tRNA^{Tyr} by Septu under conditions in which the phage natively expresses its own tRNA^{Tyr}. The evidence for partial suppression of Septu-mediated cleavage comes specifically from the plasmid-based overexpression experiment (Supplementary Fig. 4b), in which exogenous phage-derived tRNA^{Tyr} modestly reduces degradation of host tRNA^{Tyr}. We have revised the text to explicitly distinguish these experimental contexts and to clarify that the functional consequence of this partial suppression is reflected in the rescue of phage replication shown in Fig. 2b (lines 144–146).

Fig 3 - Use kb not kbp

We thank the reviewer for pointing this out. We have replaced all “kbp” labels with “kb” throughout the figures for consistency.

Extended figs - use prime symbols, not apostrophes

We thank the reviewer for the suggestion. We have revised Supplementary figure labels 3e and 4b to use proper prime symbols instead of apostrophes.

Fig 3b - very little info provided. Why is there so little impact of adding the metal, it appears already active. In extended fig 5b also provide AFold scores

We thank the reviewer for this comment. In our *in vitro* assay, the reaction buffer already contains 1 mM MgCl₂, which explains why SegB displays appreciable activity even in the “–MnCl₂” condition. We included Mn²⁺ because previous work on T4 SegB has shown that its endonuclease activity is strongest in the presence of Mn²⁺ (Brok-Volchanskaya et al., Nucleic Acids Res., 2008). Consistent with this, we observe a modest further enhancement upon MnCl₂ addition on top of the Mg²⁺-supported baseline activity. We have clarified the presence of Mg²⁺ in the reaction buffer in the figure legend (lines 1081–1083). In addition, we have added a schematic of the DNA substrate directly above the gel and, as requested, added AlphaFold3 confidence scores to Supplementary Fig. 5b.

Line 152 - state you did see growth inhibition with WT

We thank the reviewer for pointing this out. We indeed observed strong growth inhibition upon overexpression of WT SegB, and we have now stated this explicitly in the revised text, lines 202–205.

Line 209 - ladders again, please clarify

We thank the reviewer for this comment. As noted above, the ladder reflects the intrinsic instability of segmental amplification rather than incomplete plaque purification. Because amplification copy number dynamically varies among phage genomes, even plaque-purified isolates yield heterogeneous repeat numbers and therefore a ladder.

Line 233 - a pull-down showing interaction of 49.2 with OLD would be far more compelling. However, consider this only a suggestion rather than a requirement. Maybe a better route would be to KO ket surface interaction residues of 49.2 and show no longer protects?

We thank the reviewer for this helpful suggestion. We attempted to test whether Gp49.2 interacts with OLD using a pull-down assay with C-terminally tagged versions of both proteins. However, although we confirmed that the tags did not affect OLD defense activity or the inhibitory effect of Gp49.2, we were unable to detect Gp49.2 even in the input samples (**Fig. A**). We suspect that the extremely small size of Gp49.2 (~6.5 kDa) made its detection by immunoblotting technically challenging under our experimental conditions.

Figure A | Pull-down assay with functional OLD::3xFLAG and Gp49.2::c-Myc proteins

Therefore, guided by the AlphaFold3 prediction, we pursued the reviewer's suggested approach and performed structure-guided mutagenesis of Gp49.2. Based on the predicted interface with OLD, we targeted several surface-exposed residues (Met13, Thr15, and Trp32) for alanine substitution. Functional assays revealed that while the T15A and W32A single mutants retained near wild-type activity, mutation of Met13 alone (M13A) was sufficient to completely abolish the ability of Gp49.2 to neutralize OLD-mediated immunity (revised Supplementary Fig. 9c-d). We have included these new data in the revised manuscript (lines 319-329).

Reviewer #1 (Remarks to the Author):

The authors have greatly improved the manuscript in response to our suggestions. Below are a few minor points that remain:

We thank the reviewer for their positive assessment of our revised manuscript and for the constructive suggestions.

line 57 from an *E.coli* clinical strain (not the)

We thank the reviewer for pointing this out. We have corrected the wording to “an *Escherichia coli* clinical strain” to avoid implying uniqueness.

When the repeats collapse without immune pressure, is the SegB truncation is restored to the WT copy? please add this information when discussing the collapse.

We thank the reviewer for raising this point. Upon passaging in the absence of Septu Ec^{B88}, the ladder-like amplified products collapse and the population converges to a predominant band of approximately the wild-type size (Fig. 3c). We have clarified in the text that we did not sequence this collapsed product; therefore, while its size is consistent with the wild-type configuration, we cannot exclude alternative rearrangements yielding a similar-sized product.

Line 190 sounds as if nuclease activity is necessary for viral propagation, it should be specified that it is indispensable for segmental amplification of the locus of interest *in vivo*.

We thank the reviewer for this clarification. We have revised the text to specify that the nuclease activity of SegB is indispensable for segmental amplification of the *segB* locus *in vivo*, rather than for viral propagation per se.

Line 217 - reproducible cleavage sites implies replicates of the experiment were performed, please include this information in the legend

We thank the reviewer for this suggestion. We have added a sentence to the figure legend specifying that reproducible cleavage sites were defined based on concordance across independent SITE-seq replicates (n = 3).

Line 227 - they claim that motif-disrupted substrates showed little to no cleavage in Fig 3h; while the 30nt mut substrates appears more abundant, cleavage products are still present

and appear to my eye equal to the Conc. motif. If claiming cleavage efficiency, quantification of replicate assays should be provided.

We thank the reviewer for this comment. We agree that cleavage products are still detectable for the motif-disrupted substrates in Fig. 3h. To address this point, we have revised the text to avoid overstating the reduction in cleavage efficiency and have now included quantitative analysis of cleavage efficiency from independent replicate assays. This analysis shows that substrates containing the consensus motif exhibit consistently higher SegB-dependent cleavage compared to motif-disrupted substrates, while cleavage of the mutant substrates remains clearly detectable under these in vitro conditions.

Line 327: misleading; Gp49.2M13A still functions for T4.

We thank the reviewer for pointing this out. We have revised the text to clarify that the M13A substitution strongly reduces Gp49.2-mediated inhibition of OLD in T2 and T6, while retaining partial inhibitory activity in the context of T4.

Reviewer #2 (Remarks to the Author):

The authors have done a thorough job of addressing my concerns. I congratulate them on a lovely story. No further suggestions from me.

We thank the reviewer for the positive evaluation of our manuscript and for their constructive feedback throughout the review process.